# Localized Symbolic Knowledge Distillation for Visual Commonsense Models

**Jae Sung Park**[1], **Jack Hessel**[2], **Khyathi Chandu**[2],
**Paul Pu Liang**[2,4], **Ximing Lu**[1,2], **Peter West**[1,2],
**Youngjae Yu**[5], **Qiuyuan Huang**[3], **Jianfeng Gao**[3], **Ali Farhadi**[1,2], **Yejin Choi**[1,2]

[1]University of Washington [2]Allen Institute for Artificial Intelligence [3] Microsoft Research
[4]Carnegie Mellon University [5]Yonsei University

## Abstract

Instruction following vision-language (VL) models offer a flexible interface that supports a broad range of multimodal tasks in a zero-shot fashion. However, interfaces that operate on full images do not directly enable the user to "point to" and access specific regions within images. This capability is important not only to support reference-grounded VL benchmarks, but also, for practical applications that require precise *within-image* reasoning. We build Localized Visual Commonsense models, which allow users to specify (multiple) regions-as-input. We train our model by sampling localized commonsense knowledge from a large language model (LLM): specifically, we prompt a LLM to collect commonsense knowledge given a *global* literal image description and a *local* literal region description automatically generated by a set of VL models. With a separately trained critic model that selects high-quality examples, we find that training on the localized commonsense corpus can successfully distill existing VL models to support a reference-as-input interface. Empirical results and human evaluations in a zero-shot set up demonstrate that our distillation method results in more precise VL models of reasoning compared to a baseline of passing a generated referring expression to an LLM [1].

## 1 Introduction

Large language models are capable of efficiently performing a wide array of tasks in a zero-shot fashion. For text-only models, one commonly adopted interface is a flexible, language specification of inputs coupled with an imperative request, *e.g.*, "`[article text]. Summarize this article.`" Similarly, a natural extension allowing visual inputs manifests as, *e.g.*, "`[image]. Describe this image`".

But, as models expand beyond text-only modalities, they should incorporate more flexible forms of user input as well. Allowing users to specify individual objects/actors/regions within an image as part of the input query is an important challenge, e.g., the `[image]` `[request]` interface above would not directly a user to ask `Why is [this person in the image] sad?`. One option would to simply require users specifically describe the piece of the image they are attempting to specify, *e.g.*, "`[image]` `[description of specific region]` `[request]`". But authoring concrete referring expressions is not only cumbersome, particularly for scenes with lots of objects (e.g., "the person in the red jacket on the left of the scene with their arms crossed") but also challenging, even for humans: [11] argue that a good referring expression should both specify the reference precisely, but also, follow Grice's maxim of Quantity, i.e., provide no extra information. Given this tension, many

---

[1]Code will be released in `https://github.com/jamespark3922/lskd`

popular referring expression datasets are gathered in a sophisticated "gamified" fashion [53, 22], which aims to balance underspecification vs. verbosity.

We argue instead that users of vision-augmented LLMs should instead be able to pass localized visual references simply by "pointing" to regions within the image [4, 48, 40]. This enables models to focus on the region while interpreting the user's request in a more intuitive fashion, and provide more accurate and contextually relevant responses. By incorporating localized visual references, the model can better understand and interpret complex scenes, thereby improving its performance on tasks requiring a detailed understanding of the visual context.

We propose Localized Symbolic Knowledge Distillation (LSKD): the core idea is to provide literal descriptions of images to a large language model, and allow that model to connect-the-dots between these literal descriptors (e.g., lists of objects) and a holistic perspective of the scene. Different from recent works which also distill from an LLM conditioned on visual descriptors symbolically [34, 74], we additionally provide a localized reference to a particular region within the image and design prompts to encourage the LLM to generate commonsense inference about that specific region. After sampling, we train Localized Visual Commonsense models to generate commonsense triples conditioned on the image and the region directly; we show that this process effectively distills the LLM's capacity for global+local scene understanding highlighted by zero-shot results on localized visual reasoning benchmarks and human evaluation.

In summary, our main contributions are:

1. A new scalable framework that can generate reliable and localized visual commonsense statements.
2. *The Localized Commonsense Knowledge Corpus*: 1M localized commonsense inferences posed over 250K images. This dataset can be used to expand the capacity of existing vision+language models to incorporate references-as-input with no architectural modifications.
3. Achieving the SoTA zero-shot performance for three localized visual reasoning tasks.
4. Human evaluation results suggesting that a strong student model outperforms the teacher model in answering localized visual commonsense questions.

## 2 Distilling Localized Visual Commonsense from a LLM

Here, we describe our LSKD pipeline to distill visual commonsense from a LLM. Prior works have explored powerful LLM as the teacher model (GPT-3, ChatGPT) to apply knowledge distillation for language-only reasoning tasks [58, 33, 3]. Multimodal inputs offer additional challenges in grounding regions to relevant texts. Our work addresses this challenge by automatically generating reliable and diverse knowledge statements for multimodal input, to further reason about regions within an image.

Figure 1 shows the overall framework of LSKD[2]. To learn from the LLM as our teacher model, we verbalize the image into a set of dense text statements generated by global descriptors that provide relevant, general overall semantics of the image, and local descriptors that talk about specific regions in the image. We then pass these automatically generated descriptions to LLM and prompt to mine localized, commonsense statements about the image at scale (See the Appendix for the exact prompt).

As LLMs comprehend multimodal input only through machine-generated image-to-text verbalization, they are prone to hallucination and generation of inconsistent statements about the image. For instance, an incorrect verbalizer output, as in Figure 1, might cause the LLM to produce visually incoherent statements like "[1] is holding a surfboard". To minimize errors in modality translation, we construct a critic model, trained on a limited set of high-quality, hand-annotated instances to detect and remove such inconsistencies. This critic model mimics human judgment in evaluating the generated commonsense knowledge, so that we can intentionally oversample localized knowledge data, and utilize it to filter out non-relevant instances. Finally, we finetune a vision-language model on the high-quality synthetic data to facilitate zero-shot localized visual commonsense reasoning. We use 250K images in union of Visual Genome [26] and VCR [66], which include a diverse set of social situations involving people and objects, as the seed images to collect the knowledge corpus. After filtering, we collect 1M instances of Localized Commonsense Knowledge Corpus with information grounded to specific regions in the image (see Appendix A for more details).

---

[2] For visualization purposes, we provide a shortened version of verbalizations. The full verbalization output with the prompt to call ChatGPT is shown in the Appendix.

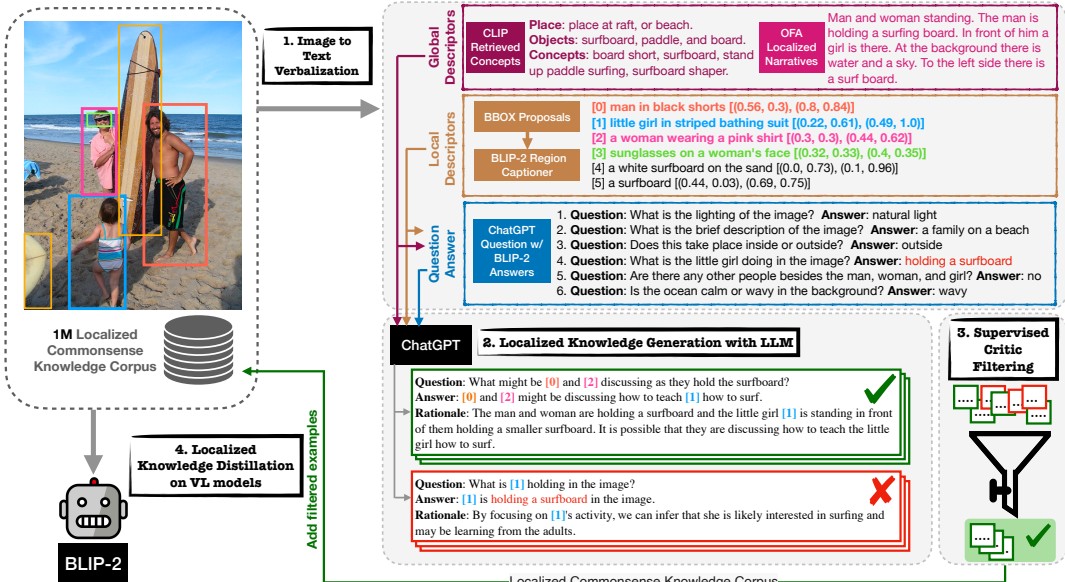

Figure 1: Pipeline of our LSKD framework. 1) Diverse vision-language descriptors are used to verbalize images. 2) LLMs leverage the global and local descriptors to generate grounded commonsene knowledge. 3) We annotate a small subset of data to train a supervised critic model that can filter instances displaying incorrect visual details or incoherent reasoning. The critic model filters the rest of generated statements to finalize the data pool. 4) A multimodal model is finetuned on the synthetic data to support localized visual commonsense reasoning in a zero-shot manner.

## 2.1 Image to Text Verbalization

We first describe our methods for verbalizing (i.e., writing out in natural language) images/regions to text. Note that this step does not require images with text annotations for the target datasets, unlike prior work [34], and can be applied to any set of images. We focus on deriving *global* image descriptions, *local* region descriptions, and *dynamic* question-answer pairs for each image. Figure 1 gives a schematic of our process which includes an example image verbalization.

**Global descriptors: Image Verbalizers** Following [69], we use the CLIP-ViTL model in a zero-shot fashion to extract basic concept information about the image using a template. We retrieve places from the Place365 [71], objects from TencentML-Images [59], and concepts from OpenImages [27] to arrive at global concepts. Specifically: we use the top 2 places, the top 3 object labels, and the top 3 concepts. In addition to concepts, we also get narrative descriptions of the entire image. For this, we fine-tuned OFA-Huge [54] on the Localized Narratives [44] corpus, which pairs 849K images with multi-sentence descriptions (we ignore the mouse trace information and simply treat the task as image-to-text captioning). We sample 5 localized narratives for each image using a temperature of 1.1.

**Local descriptors: Region Verbalizers.** Global descriptors alone often fail to capture the intricate details of specific regions within an image, leading to a potential bottleneck in understanding scenes with more visual precision and enabling localized reasoning. We employ local descriptors that provide more grounded visual statements. To do so, we sample bounding box regions for the image using region proposal models from object detection literature [32]. We then train a region captioning model that maps from (image, region) → description of the region. We fine-tuned the generative version of BLIP-2 [29] with the FLAN-t5-xxl [7] backbone. We trained on datasets that provide descriptions of regions within images. a combination of RefCOCO/RefCOCO+/RefCOCOg [64, 37], Sherlock Clues-only [19] (277K), and VisualGenome [26] (1.96M): all of these datasets provide descriptions of given regions within images. Following [68, 62], we render the bounding box in the image itself to allow the model access to the bounding box's location. More details of the local descriptors are in Appendix E.

| QA MSE | Rationale MSE | Precision | Recall | F1 |
|:---:|:---:|:---:|:---:|:---:|
| | | 64.7 | 64.2 | 64.3 |
| ✓ | | 66.3 | 65.2 | 65.7 |
| | ✓ | 66.0 | 64.3 | 64.8 |
| ✓ | ✓ | **66.8** | **65.7** | **66.0** |

Table 1: Analysis of BLIP-2 based critic model. We see that adding the multi-class regression loss further improves the performance of critic model.

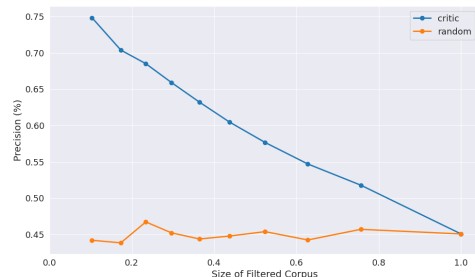

Figure 2: Precision of Critic Model with varying threshold values to filter the corpus size. Precision is increased significantly by using the supervised critic model to filter the corpus.

**Dynamic descriptors: Q+A Verbalizers** Finally, to support a holistic understanding and enable models to dynamically probe for potentially missing context, we acquire more fine-grained details about the scene using a series of questions and answers. Following [73], we prompt an LLM to ask short, simple questions conditioning on the global and local descriptors as context, and query BLIP-2 [29] in a zero-shot fashion to answer the questions. We specifically collect 15 question/answer pairs for each image.

## 2.2 Localized Commonsense Knowledge Generation

For all experiments, we use ChatGPT as our LLM,[3] though in principle, any instruction-tuned LLM could be used. We use question-answering-rationale (QAR) for knowledge representations. QAR representations are flexible, and have been successfully adopted as meaning representations in areas ranging from formal semantics [17, 38, 24] to commonsense reasoning [50, 66].

Given the verbalization of images, we prompt ChatGPT to come up with an interesting and complex question with the possible answer that requires rationale to justify the reasoning. We support two versions of localized knowledge generation. One that refers to specific regions in the image either by their assigned numerical IDs and bounding box coordinates (*e.g.* `What is [2] doing in the image?`) for more precise localization, and one that uses descriptive phrases (*e.g.* `What is [the woman wearing a pink shirt] doing in the image?`) for more contextual and detailed references. Qualitatively, we observe that the LLM is able to connect the IDs and the region descriptions successfully, and create a convincing set of localized commonsense knowledge corpus. For each image, we prompt ChatGPT three times to generate three unique QARs sequentially. We do this for ID-based and description-based references (see Appendix for the prompts), and collect 18 localized instances per image.

## 2.3 Training the Critic Model

We train a supervised critic model to reflect the human acceptability of generated data. We allocate a subset of 20K statements to train the critic model, and 4k for evaluation. The "accepted" instances should generally deliver the visually correct information and exhibit coherent reasoning. For each QAR, we ask two human annotators to rate from 1 to 3 (reject / maybe / accept) if 1) the QA displays visually correct information (QA rating), and 2) the rationale justifies the answer while being aligned with the image (QA → R rating)[4]. We then assign binary label if at least one annotator has included reject for any of the two rating criteria. Using this labeling scheme, we found that only 45% of the instances are labeled as accepted, suggesting that aggressive filtering by the critic model is required.

For the model, we use a stage-1 pre-trained BLIP2 [29] with ViT-G [12] image encoder to do the critique. Following their finetuning scheme on retrieval tasks, we train the image encoder and Q-Former together, not freezing the weights. We add a linear layer to the image-text matching head that has been pre-trained to capture the multimodal content, and train it to perform the classification.

---

[3] https://openai.com/blog/chatgpt
[4] The second criterion is automatically rejected if the QA has already rejected in the first pass

| Descriptors Used | Average Critic Score |
|---|---|
| Full Descriptors | 58.4 |
| (-) CLIP Concepts | 52.1 |
| (-) Localized Narratives | 56.1 |
| (-) Global Descriptors | 54.3 |
| (-) Local Descriptors | 49.8 |
| (-) QAs | 49.0 |

Table 2: Ablation study of the descriptors. We remove one of the descriptors from full descriptors when calling ChatGPT to generate the corpus, and calculate the average critic score to rate the generations (higher the better).

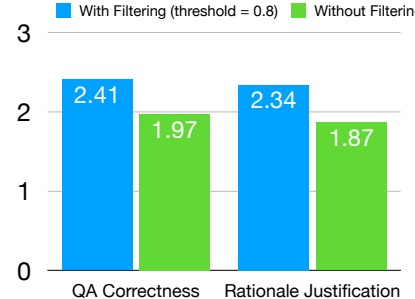

Figure 3: Human judgment of corpus with and without filtering. We get the average ratings in Likert scale (from 1 to 3) from three human annotators.

We utilize the two rating criteria (QA and QA → R) to further inform the critic model to know what caused the humans to reject the QARs. We achieve this by multi-task training of critic model. The ratings containing reject are given the regression label of 0, while the average of two QA and QA → R ratings is calculated to get the regression label $y_{QA}$ and $y_{QA \to R}$. Along with the binary classification loss, the image-text matching head is further trained with mean squared error (MSE) losses with $y_{QA}$ and $y_{QA \to R}$. Table 1 shows the performance of critic model on the above train and eval split. We empirically see that adding the multi-task loss (QS MSE and Rationale MSE) further helps the performance of classification.

**Analysis of Supervised Critic**   How reliable is the critic model on filtering erroneous instances? In the annotation stage, we have observed that only 45% of the instances would be considered as valid by humans. We explore tuning different thresholds of critic model to filter the data (*e.g.* keep instances whose predicted scores are higher than the threshold), and see if higher acceptability can be achieved with higher threshold. Figure 2 shows a plot of precision value (instances labeled as "accept") by the filtered corpus size. We see a consistent trend where removing the corpus with more critical criteria yields higher acceptability. Specifically, it jumps from 45% of 70% acceptance if 20% are maintained by the critic model. We use this threshold value of 0.8 to apply the critic model. Note that filtering the corpus randomly, on the other hand, doesn't have any influence on the acceptability.

In addition, we run human evaluation to measure the acceptability of data with and without filtering. We collect 500 instances the same way critic model labels are collected: 1) is the QA visually correct? and 2) does the rationale justify the answer? Likert scores from [1-3] are calculated for each criteria (higher the better). Figure 3 shows the human evaluation results, and we see that the dataset with filtering is more favored by humans than without filtering.

**Are All the Descriptors Necessary?**   We run ablation studies of the descriptor components in the ChatGPT prompt and use the critic model to score the ChatGPT generations. We collect QAR instances for 500 images and calculate the average critic score, with higher score aligned with human preference. Table 2 shows the result when one of the descriptors is removed from full verbalizations. We see that using all descriptors provides the best results, and in fact the QA descriptor provides the biggest jump (from 49.0 to 58.4).

## 2.4   Training with the Localized Corpus

We explore the distillation of localized commonsense knowledge by finetuning discriminative and generative vision language model on our corpus. For the corpus that mentions IDs and bounding box coordinates, we follow [68, 62, 67, 19] by directly drawing colored highlights around the regions in the images where the region IDs and highlights are consistent throughout the corpus (*e.g.* [0] always gets the color pink).

During training, we additionally apply region-based augmentation by reassigning the IDs with a random order while still keeping a consistent color coding (*e.g. What might be [0] and [1] discussing?*

→ *What might be [1] and [3] discussing?*). We similarly vary the number of regions to be shown in the image, while ensuring that the mentioned IDs are drawn in the image. With these tricks, the modifications are performed in the input image and text to enable localization, while the architecture and training objectives of the vision-language model remain unchanged.

We use the BLIP-2 [29] as the vision and language backbone model. Given the recent success and efficiency of visual instruction methods, [34, 74, 29, 10], we freeze the weights of visual and language model and only train the Qformer [34] learns to map visual to text tokens. For discriminative tasks, we apply the stage 1 pre-training objective with Image-Text Contrastive, Image-Text Matching, and Image-Text Grounding Losses. We further explore generative performance with the FlanT5$_{XXL}$ [57] language model and Mini-GPT4 that tunes the Vicuna-13b-v0 language model [6, 52] to understand visual tokens. We refer to [29] for more training details.

## 3  Experiments & Results

We use the OpenAI Chat API with gpt-3.5-tubro engine and a temperature of 0.8 to prompt the LLM to collect knowledge data. The BLIP-2 critic model is trained with total batch size of 256, learning rate of 1e-5, max 10 epochs. The visual encoder (ViT-G) model is additionally trained instead of kept it as frozen.

The discriminative BLIP2 is trained with 256 batch size and 128 max sequence length for 1e4 iterations. The BLIP-2 FlanT5$_{XXL}$ and Mini-GPT4 models are trained with 64 batch size and 2e4 iterations. All models are trained with learning rate of 1e-5, Adam optimizer [23], linear warmup with cosine annealing, and image size of 480 using 80GB 4 A100 GPUS. We do not finetune the ViT or the language model, and only train the QFormer shown by the success from prior work [29, 10, 34].

### 3.1  Downstream Tasks

**Localized Visual Commonsense Reasoning**  We evaluate on a set of visual commonsense reasoning tasks that involve identifying and referring specific regions in the image in a *zero-shot* setting. VCR [66] is a task that requires choosing the right answers for question (Q → A), and rationales justifying the answer (QA→ R) from four multiple choice options. The results are combined with (Q → AR) metric that requires selecting the right answer and rationale. VisualCOMET [41] is a commonsense knowledge graph of understanding specific people's intent, and what they would do before and after, and adopt their Acc@50 task of retrieving ground truth inferences from 50 candidates . Sherlock [19] is a visual abductive dataset that includes the comparison evaluation of ranking of 10 text inference candidates aligned with human preference. All the aligned tasks require reasoning about specific regions or people in the image, and getting the image-text similarity score from a model.

**Non-Localized Visual Reasoning**  We measure the effectiveness of the localized knowledge corpus on other vision-language tasks not limited to datasets with no bounding box annotations. We specifically focus on ones that require high-level reasoning that would benefit from visual commonsense corpus. AOKVQA [47] requires outside world-knowledge to answer questions and we evaluate on their multiple choice setting. SNLI-VE [61] is an inference based visual entailment that tests fine-grained image understanding. The task is to predict whether the image semantically entails the text, and specifically classify if the image-text is one of {entailment, neutral, contradiction}. Visual7W [75] is visual QA with focus on visual grounding, and we evaluate on the subset of telling questions that have textual answers (Telling QA).

**Baseline models**  We include CLIP as our baseline as it has shown strong zero-shot generalization results for various image-text alignment tasks [45]. Following [56], we exclude the question in the text input and acquire the image-text cosine similarity score to do the task. CLIP-Event is a CLIP model pre-trained on their VOA dataset crawled from news websites [31]. BLIP is image-text alignment model trained with additional generation objective and boostrapped image captions [30]. We lastly evaluate the zero shot performance of BLIP-2 [29] varying the visual encoders before applying knowledge distillation. We do not draw bounding boxes in the image nor include id tags in the text description, as these models have not been pre-trained in this domain.

| Approach | Localized | | | | | Non-Localized | | |
|---|---|---|---|---|---|---|---|---|
| | **VCR** | | | **Sherlock** | **VisualCOMET** | **AOKVQA** | **SNLI-VE** | **Visual 7w** |
| | Q → A | QA → R | Q → AR | Comparison | Acc@50 | Mult. Choice | Classification | Telling QA |
| CLIP-Event [31] | 52.4 | 49.2 | - | - | 22.4 | - | - | - |
| CLIP ViT-B-16* [45] | 54.8 | 48.6 | 26.6 | 9.9 | 33.0 | 58.3 | **36.0** | 65.9 |
| CLIP ViT-L-14x336 [45] | **56.3** | **51.3** | **29.9** | 10.9 | 34.8 | 61.0 | 31.9 | 66.7 |
| BLIP ViT-L [30] | 47.2 | 42.5 | 20.1 | 18.6 | 31.3 | 61.3 | 34.2 | 69.4 |
| BLIP-2 ViT-L [29] | 52.3 | 48.1 | 25.3 | 18.7 | 36.7 | 65.0 | 31.7 | 73.6 |
| BLIP-2 ViT-G [29] | 56.1 | 49.8 | 28.0 | **19.5** | **39.0** | **68.0** | 33.4 | 77.1 |
| BLIP-2 ViT-G + LSKD | **59.0** | **56.4** | **33.4** | 29.7 | 40.3 | 68.9 | 40.3 | 79.5 |

Table 3: Zero-shot results on the localized and non-localized visual reasoning tasks. *Zero shot VCR results directly obtained from [56]. For CLIP, we follow [56] by omitting the question and having the answer (with rationale) as text input to calculate the image-text similarity. For BLIP-2, we maintain the question text input as it improves the performance.

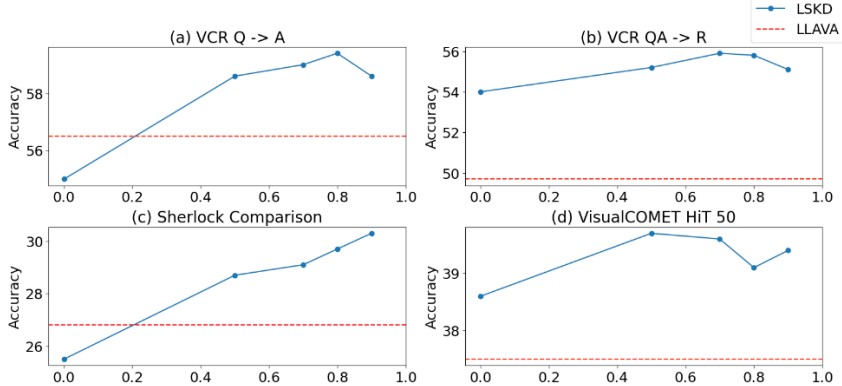

Figure 4: Effect of data quality controlled by filtering threshold on different datasets. The x-axis shows the threshold for filtering and the y-axis is the accuracy metric in percentage. We compare training our model on the LLaVA-instruct dataset (red) and ours (blue).

## 3.2 Zero-Shot Visual reasoning results

Table 3 shows the zero-shot results on the downstream tasks. For localized reasoning tasks, we first observe that scaling the visual encoder size (CLIP ViTB-16 vs ViT-L-14x336; BLIP-2 ViT-L vs ViT-G) in general improves the performance. CLIP outperforms BLIP-2 on VCR tasks but fall short on Shlerock and VisualCOMET. After applying localized symbolic knowledge distillation (LSKD) to BLIP-2, there is a consistent improvement over the BLIP-2 model on all downstream tasks (5.4% on VCR Q → AR, 10.2 on Sherlock Comparison, 1.3% on VisualCOMET Acc@50).

For non-localized reasoning tasks, we observe a similar pattern. Interestingly, applying LSKD improves the performance of BLIP2 model further across all the tasks (AOKVQA, SNLI-VE, Visual7W) over the vanilla model, despite these tasks not being the primary target domain. This demonstrates that the advantages of distilling models with localized reasoning can be transferred to high-level visual commonsense tasks, thanks to the visual precision and enhanced reasoning abilities learned from the generated knowledge corpus.

**Influence of Critic Filtering on Downstream Tasks** How does the process of critic filtering influence the performance of downstream tasks? Keeping the size of the selected statements the same at ∼ 300K, we select qualified knowledge statements with varying prediction thresholds. We also compare with training on the LLaVA-instruct dataset which similarly prompts an LLM (GPT-4) to generate complex questions using ground truth verbalizers [34]. Figure 4 presents the resulting performances at these diverse thresholds across different datasets. Compared to LLaVA, we observe that localized knowledge statements without filtering does not show any improvement for the downstream model, while any thresholding over 0.2 is consistently better than LLaVA across all datasets. For tasks that demand relatively moderate commonsense, such as VCR Q→A and Sherlock Comparison, increasing the threshold consistently improves the model performance. For

|  |  |  | Localized | | | Non-Localized | | |
|---|---|---|---|---|---|---|---|---|
| | | | **VCR** | **Sherlock** | **VisualCOMET** | **AOKVQA** | **SNLI-VE** | **Visual 7w** |
| **Dataset** | **Size** | **Annotator** | Q → AR | Comparison | Acc@50 | Mult. Choice | Classification | Telling QA |
| Zero-Shot | NA | NA | 28.0 | 19.5 | 39.0 | 68.0 | 33.4 | 77.1 |
| Sherlock [19] | 300K | Human | 34.6 | 30.5 | 39.7 | 67.2 | 38.6 | 70.1 |
| VisualCOMET [41] | 1.2M | Human | 31.8 | 25.3 | 50.2 | 68.5 | 35.6 | 70.8 |
| LLAVA-Instruct [34] | 150K | GPT-4 | 28.1 | 26.9 | 37.5 | 71.0 | 42.6 | 79.5 |
| LSKD (Ours) | 150K | ChatGPT | 33.3 | 28.6 | 39.7 | 69.6 | 38.0 | 75.9 |
| LSKD (Ours) | 1M | ChatGPT | 33.4 | 29.7 | 40.3 | 68.9 | 40.3 | 79.5 |

Table 4: Ablations of BLIP-2 ViT-G trained with varying sources of visual-knowledge corpus annotated by humans and machines. We break down to visual reasoning tasks that require localized reasoning and those do not. Critic filtering is applied to the LSKD corpus (Ours).

tasks requiring a higher degree of commonsense such as VCR QA→R and VisualCOMET Hit@50, the performance increases until a certain threshold and then fluctuates. We speculate that a more grounded critic model could potentially mitigate this fluctuation, and we intend to investigate this in our future work. Overall, our findings suggest that higher thresholds (i.e., more critical filtering) tend to yield superior quality generations, thereby enhancing the performance on downstream tasks.

## 3.3 Human vs Machine Annotated Corpus

Can training on machine annotated corpus result in competitive performance with human annotations? In Table 4, we compare the performance of BLIP-2 ViT-G trained on existing human-annotated corpora with our machine-annotated corpus across various scales. First, we found that increasing the size of our training corpus (150K vs 1M) leads to consistent improvement across all tasks, indicating a promising trend of scaling law for synthetic training data. Regardless of the size, training on our dataset yields considerable benefits over the zero-shot model on localized reasoning tasks.

Next, we observe that training on human annotated corpus vastly improves the performance of their relative tasks (e.g. training on VisualCOMET boosts performance from 39.0 to 50.2). However, this can lead to inferior results on other visual reasoning tasks than the zero-shot counterpart. For instance, the performance on Visual7W drops from 77.1 (Zero-shot) to 70.1 (Sherlock) and 70.8 (VisualCOMET). This suggests that human-designed datasets may limit task generalization due to their lack of diversity. Interestingly, we see that training the model our full LSKD corpus (1M) leads to uniform gains over the zero-shot model across the tasks, and even outperforms the human annotation corpus for the non-localized tasks as well. This shows that machine-annotated datasets, when curated and scaled adequately, can indeed rival or even surpass the performance of models trained on human-annotated corpora.

We directly compare training on ours and the LLaVA dataset. Regardless of our dataset scale, we observe that LSKD + filtering wins over training on the LLaVA corpus on localized reasoning benchmarks, even when using a less powerful teacher model (ChatGPT vs GPT-4). This suggests that our creation of a new localization corpus is crucial to support the model with grounded reasoning. On the other hand, LLAVA wins on non-localized reasoning tasks as they are aligned with the nature of training corpus. We thus observe that the appropriate application of the corpus can be task-dependent, and adopting a selective approach towards generating the corpus may result in significantly enhanced performance across various benchmarks.

## 3.4 Localized Reasoning with Generative Models

We extend LSKD to train generative models that can refer and talk about highlighted regions in image. We finetune BLIP-2 FlanT5 and Mini-GPT4 and prompt them to answer questions from the VCR data. As there is no baseline zero-shot model that can reason about regions to answer questions, we make a direct comparison of the student LSKD model to the teacher LLM with access to verbalizations. We ask annotators on Amazon Mechanical Turk (AMT) platform to run head-to-head comparisons (with ties) on three criteria, if the answer delivers: 1) visually correct details, 2) informative and interesting information, and 3) content that sounds plausible. Finally, they select their overall preference. We take the majority vote of 3 annotators, and disregard the instance if there is no clear majority.

| Model | Correctness | Informativeness | Plausibility | Overall |
|---|---|---|---|---|
| ChatGPT w/ Vebalizers | 34.7 | 33.9 | 39.6 | 45.0 |
| BLIP-2 (FlanT5$_{XXL}$-11B) + LSKD | 31.7 | 41.0 | 30.2 | 41.2 |
| Tie | 33.7 | 25.1 | 30.2 | 13.1 |
| ChatGPT w/ Vebalizers | 29.8 | 31.7 | 36.8 | 40.6 |
| Mini-GPT4 (Vicuna-13B) + LSKD | 34.3 | 53.0 | 34.2 | 49.1 |
| Tie | 35.9 | 15.3 | 30.0 | 10.3 |

Table 5: Human evaluation of generative models with LSKD vs Chat-GPT with verbalizers. Humans are asked to choose the better generation or tie if they share the same quality.

Table 5 shows the human evaluation results. We observe that the LSKD generally wins in informativeness over ChatGPT, but not in plausibility. We see a conflicting pattern in correctness and overall preference, where Mini-GPT4 is equipped with a more powerful language model that outperforms the teacher model while BLIP-2 falls short. Unlike previous language-based distillation where a relatively weak student model can outperform the teacher [58, 3], we see that a strong student model may be required to outperform the teacher LLM in the multimodal domain.

**Qualitative Results**    Figure 5 presents a comparative analysis of question-answering with rationale results on VCR samples generated by ChatGPT, LLaVA [34] and `Ours`. Both Ground Truth (GT) and `Ours` consistently identify the correct entities, with `Ours` model often providing broader context, which is uncertain on rare occasions (*e.g.* "likely the bride"). On the other hand, ChatGPT predominantly focuses on observable actions or states as described in the text context, occasionally resulting in the misidentification of the visual entities and their relations. In the third example in Figure 5, "waiting for someone" focuses on the observable state "standing still", missing visual detail such as a cave, holding a flame, and surrounding context. LLaVA, in contrast, generally provided a broad overview while identifying a specific visual entity in most cases. However, it often struggled to accurately identify specific entities within the complex scene (*e.g.* "holding a wine glass" in Figure 5.(1), "cigarette" in Figure 5.(3) ). Compare to LLaVA, `Ours` often aligned closely with GroundTruth and incorporated both actions and inferred knowledge in its answer. Overall, `Ours` delivers a more nuanced and contextually-rich response.

# 4   Related Work

**Knowledge Distillation**    Recent research [1] has extensively explored the use of language models as knowledge bases, highlighting their potential in reasoning, explainability, and consistency, which can enhance downstream tasks by distilling knowledge from LMs. [15] demonstrated how knowledge augmentation explicitly from knowledge bases and implicitly from GPT-3 improved open-domain multimodal tasks. [33] showed that overgeneration with GPT-3 from exemplars to filter, as well as reviewed by humans, is a new and reliable way to create an NLI dataset with human and AI collaboration. This setup also has the advantage of bringing forth cultural internalization via human collaboration [9]. Previous works have explored knowledge distillation in the multimodal domain by prompting the teacher LLM with human-annotated verbalizations [34, 74, 10]. Our work is different in that it generated *localized* commonsense descriptions and the knowledge generation framework can operate a scale without the need for aligned descriptions.

**Filtering**    [2] filters the generated sentences using a classifier trained on original sentences and a set of generated sentences. [55] used the same technique to filter out synthetic data created, which is of low quality. Large language models can be used to refine the commonsense knowledge retrieved from web contents by filtering the data generated from these models [39]. They perform a consolidation step that filters topical and relevant assertions based on OpenIE.

**Multimodal commonsense reasoning**    requires more profound real-world knowledge, potentially spanning logical, causal, and temporal relationships between concepts. For example, elements of causal reasoning are required to answer the questions regarding images in VCR [66] and Visual-COMET [41], while other works have also introduced datasets with video and text inputs to test for temporal reasoning (*e.g.*, Social-IQ [65], MovieQA [51], MovieFIB [36], TVQA [28]). Benchmarks for multimodal commonsense typically require leveraging external knowledge from knowledge bases [49] or pretraining paradigms on large-scale datasets [35, 68].

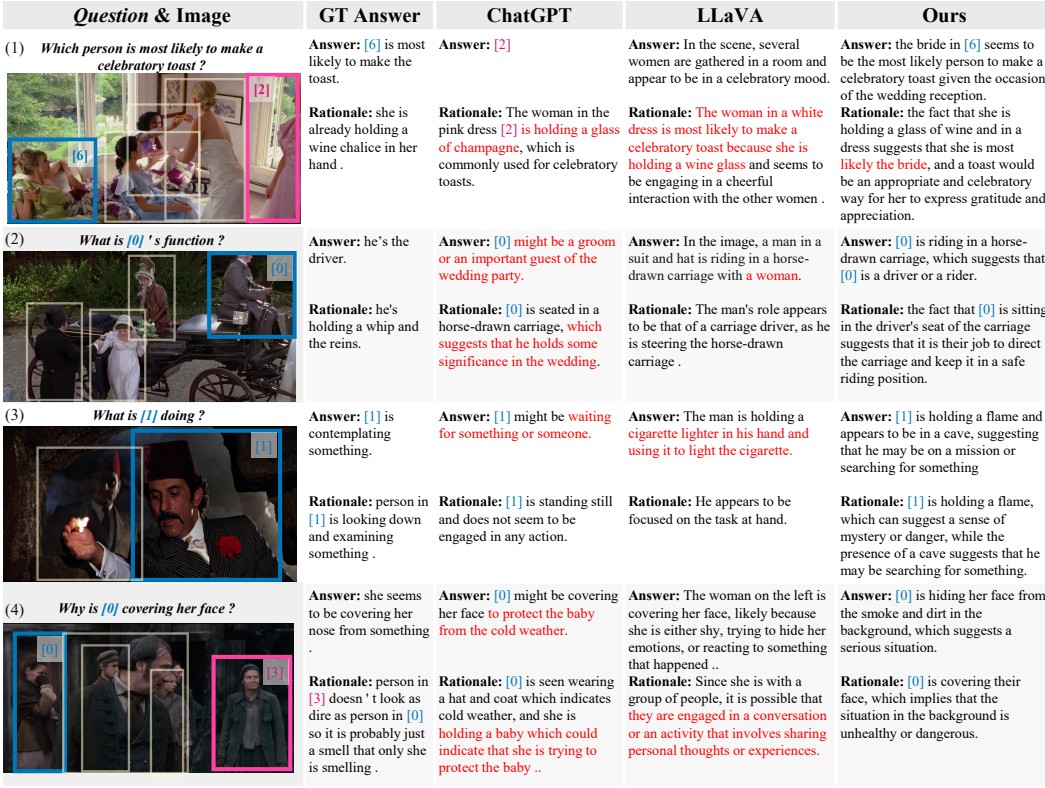

Figure 5: Qualitative examples comparing ChatGPT (the teacher model), LLAVA trained for complex visual reasoning [34], and ours. Each person referenced in the question has been assigned a unique number with a bounding box and their mention has been marked with a corresponding color. Any errors within the generated results are emphasized with a red highlight.

**Region Understanding with Multimodal alignment** Capturing elements across modalities that have a common meaning and is exemplified by tasks such as visual coreference resolution [25, 42], visual referring expression recognition [8], multimodal question answering [19, 66], and cross-modal retrieval [13, 43]. Alignment between modalities is challenging since it may depend on long-range dependencies, involves ambiguous segmentation (*e.g.*, words or utterances), and could be either one-to-one, many-to-many, or not exist at all. Resources for fine-grained alignment include Visual Genome [26] and dense captioning [21], diverse reasoning [63]. Recent methods have adopted either generative or retrieval-based methods for alignment: generative methods create localized verbalizations of the region of interest [72, 14, 21, 70], while retrieval aims to select the most accurate caption for the region of interest despite possibly given only coarse-grained paired data of captions for entire images [5, 18].

## 5 Conclusion

We present LSKD, a method for sampling localized commonsense knowledge from a large language model. With the help of supervised critic model aligned with human judgements, we create a diverse, reliable 1M localized commmonsense corpus. Training on the resulting corpus supports models that can accept region references as input, which allows users to interact with specific parts of images by "pointing;" all without having to write out a referring expression explicitly. We show that training on our corpus improves the zero-shot performance of vision-language models for tasks requiring regions-as-input. Making the critic model more critical by strict thresholding improved performance further. We present a state of the art zero-short performance with our approach opening avenues for visual commonsense models with our localized commonsense knowledge corpus.

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
