# Supplementary Material

## A    The Localized Commonsense Knowledge Corpus

Table 1 shows the detailed statistics of the corpus. We break down the corpus where the regions are referenced by their IDs and by their region descriptions. The maximum number of mentioned region IDs in the QAR is limited to 5. Figure 1 illustrates the distribution of the number of IDs.

We show the category of question types and examples in Table 2. Upon manual inspection of the corpus, we have identified specific question types that exhibit prominent characteristics. These types are associated with a collection of n-gram patterns, and questions sharing these n-grams are categorized accordingly (e.g., questions containing the terms "purpose" and "significance" are assigned to the Purpose category). Lastly, the word clouds for question, answer, and rationale are shown in Figure 2.

|  | With Region ID's | With Region Descriptions | Total Corpus |
|---|---|---|---|
| # of Images | 128,564 | 125,524 | 168,996 |
| # of QARs | 513,223 | 467,658 | 1,023,807 |
| Average # of Qs per Image | 3.99 | 3.73 | 3.86 |
| Average Q Length | 13.0 | 10.9 | 11.8 |
| Average A Length | 14.4 | 10.5 | 12.3 |
| Average R Length | 25.8 | 22.8 | 24.1 |
| Average # of mentioned ID's | 0 | 1.25 | 0.57 |

Table 1: Detailed statistics of the Localized Commonsense Knowledge Corpus.

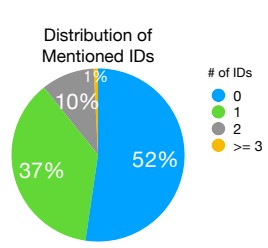

Figure 1: Distribution of mentioned Region ID's.

| Question Type | Freq (%) | Example |
|---|---|---|
| Purpose | 20.0 | *What is the purpose, What is the significance...* |
| Relationship | 10.5 | *What is the relationship, How are they related...* |
| Type | 10.1 | *What kind of, What is the type of...* |
| Emotion | 8.4 | *What emotion, What might be the feeling of...* |
| Scene | 7.7 | *Where, What time, What situation...* |
| Attribute | 7.4 | *What state, What condition, What color...* |
| Action | 5.9 | *What activity, What event, What are they doing...* |
| Inference | 5.3 | *What can you infer, What would likely, How might...* |
| Reason | 5.1 | *Why, What is the intention...* |
| Role | 4.7 | *What is the role, What is the occupation...* |
| Focus | 4.5 | *What is the main focus, What stands out...* |
| Ambiance | 4.4 | *What atmosphere, What is the mood, What vibe...* |
| Factual | 3.5 | *Is/Are there..., Do you think...* |
| Others | 2.6 | - |

Table 2: Types of questions and their examples in the corpus. To identify these question types, we manually construct a set of n-gram patterns and categorize questions based on their inclusion of these specific n-grams.

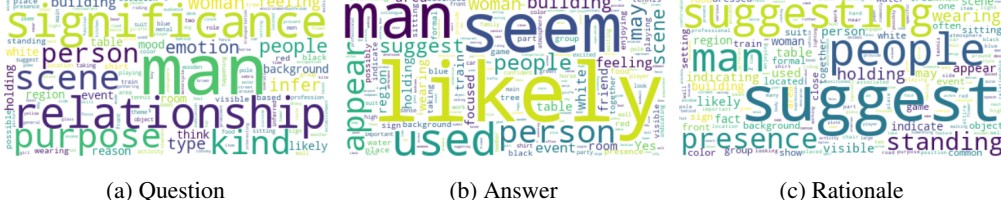

(a) Question          (b) Answer          (c) Rationale

Figure 2: Word Clouds of Question, Answer, and Rationale

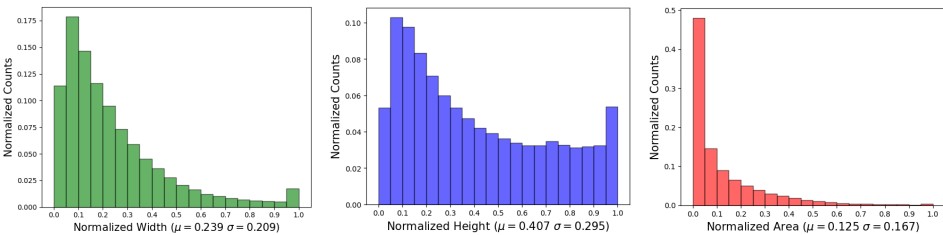

Figure 3: Distribution of bounding box sizes in the generated corpus. x-axis is the normalized box width, height, and area from 0 to 1. y-axis is the normalized counts over total number of samples.

## B  Bounding Box Distributions and Model Performance

Figure 3 shows the distribution of normalized bounding box sizes in the filtered corpus, highlighting the width, height, and the area. We notice that almost 50% of the bounding boxes have the normalized area 0.05, suggesting that small objects are well-covered in our corpus. The height shows more uniform distribution than the width, indicating that there are more bounding boxes with smaller widths and the width mainly clusters in the range of 0.1-0.2. This reveals that the corpus contains not just large and prominent objects, but also small or narrow objects that often require attentive vision models to recognize.

We use the Sherlock comparison task [4] to study the model performance change w.r.t different bounding boxes as their dataset consists of single bounding boxes with diverse sizes. The Pearson's correlation between the input bounding box size and the comparison accuracy is $\rho = -0.12$ with p-value of 0.05.

Based on the correlation, we see that the performance is actually higher for smaller objects. One might indeed initially think that larger bounding boxes would result in better performance, as they could potentially encompass more features or more of the object of interest. We hypothesize that the negative correlation is due to the following.

- Specificity: Smaller bounding boxes quite often are more specific in identifying the target objects, thereby reducing the complexity of the region and making it easier for the model to focus and reason.
- Clutterness: Larger bounding boxes might include more "noise" or irrelevant objects/background, which could mislead the model during the reasoning process as it gets distracted by extraneous details.

## C  More Details of Corpus Generation

We show the full output of our image to text verbalization pipeline using the global, region, and question-answer descriptors in Figure 5. For concepts, we acquire the visual feature $v$ and text features for each object classes $[t_1, t_2, ...t_C]$ extracted by the CLIP-ViT-L-336 model [12], and use the nearest neighbor search by their cosine distance to select the top $k$ labels for the image. We train OFA-Huge model [14] on the Localized Narratives [11] and generate 5 descriptions with nucleus sampling [5] of $p = 0.95$. BLIP-2 trained on region captions described in Section 2.1 is used to describe the regions individually. We get the questions using ChatGPT, in which we provide the global and local descriptors as context, and call the OpenAI API with the following instruction: `Here is the context for the image: {global descriptors} \n\n {local descriptors} \n\n Now, ask fifteen interesting but simple questions that you want to ask so you can get more understanding about the image`. The zero-shot BLIP-2 answers the generated question, and the QA pairs are used as the dynamic descriptors.

To generate the Localized Commonsense Knowledge Corpus, we utilize verbalization as context and present two distinct prompts to ChatGPT. In one prompt, regions are referenced by numerical IDs, while in the other prompt, regions are described using text descriptions. The specific prompts used to invoke ChatGPT are depicted in Figure 6 and Figure 7. In the former case, instances where no IDs are mentioned in the output are filtered out, while in the latter case, instances containing any IDs in

the output are excluded. An example generated using both versions of the prompts is showcased in Figure 8.

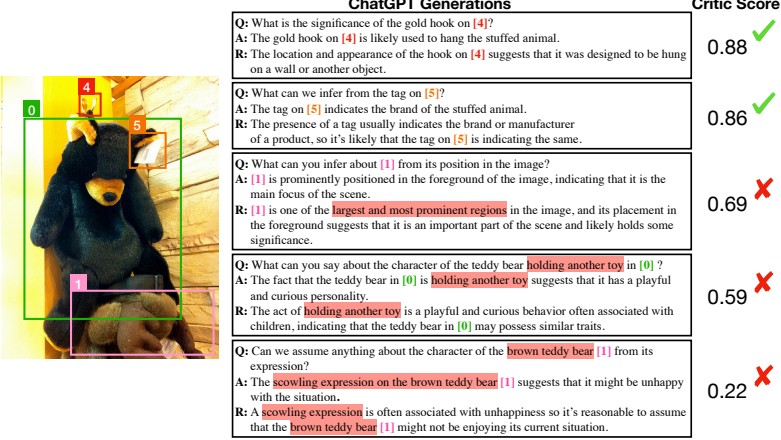

Figure 4: Qualitative examples of supervised critic filtering of ChatGPT generated data. We discard generations whose critic scores are lower than the threshold value of 0.8. Incorrect visual details are highlighted as red.

# D    Qualitative Analysis of Critic Model

Figure 4 shows qualitative examples to understand the patterns of critic model in distinguishing good and bad examples. We see that the model mostly relies on incorrect visual details (highlighted as red) lower than the correct instances. The third instance does not have glaring visual errors but are scored lower due to statement of "largest and most prominent regions", which is ambiguous but close to false. The critic model displays good calibrations in ordering the instances, such as giving the lowest score to the instance with the most visual error.

# E    More details of Local Descriptors

We train a region captioning model that maps from (image, region) → description of the region. We fine-tuned the generative version of BLIP-2 [8] with the FLAN-t5-xxl [2] backbone. We trained on a combination of RefCOCO/RefCOCO+/RefCOCOg [17, 10] (120K/80K/120K training region captions), Sherlock Clues-only [4] (277K), and VisualGenome [7] (1.96M): all of these datasets provide descriptions of given regions within images. Following [19, 16], we render the bounding box in the image itself to allow the model access to the bounding box's location.

We compared our model's captions to those generated by GRiT [15], which achieves state-of-the-art performance on dense captioning [6]. The standard evaluation metric for dense captioning combines region proposal and caption generation metrics. Because we aim to generate captions for *any* given region provided by a user (and not just model-proposed ones), we instead evaluate generation capacity *given* a region. Specifically, we conduct a pairwise human evaluation comparing the generations of GRiT on its proposed bounding boxes vs. our model's generations on the same GRiT-proposed bounding boxes. 5 authors of this work evaluated 150 randomly-sampled captioned regions from test set examples in a head-to-head setting. Annotators could select "A", "B", or "Tie": GRiT and our region captioner were randomly assigned A or B placement in a blinded fashion. Overall: while both performed well, our region captioner was preferred to GRiT on average. In 46% (347/750) of cases, annotators reported a tie, in 34% (248/750) of cases, annotators reported ours was better, and in 19% (145/750) of cases, GRiT was rated as better.

Given the (image, region) → description model, we next sample candidate regions of interest; in § **??**, we condition on these regions for the generation of commonsense knowledge. We use the ViT-H Cascade Mask R-CNN [9] trained on LVIS [3] for an initial proposal set. The detector outputs up to 300 candidate objects per image, many of which overlap or cover background objects that are not

the focus of the scene. For each image's list of candidate objects, we heuristically downsample to a set of "most interesting" regions by: 1) selecting the at-most $k = 4$ largest/most central people; 2) keeping the most central/large objects; 3) over-sampling rarer objects according to prior frequency of detection in the LVIS vocabulary; 4) limiting the number of objects of a single type per-image; and 5) downsampling overlapping region proposals to encourage broader coverage of the pixel area of the image.

## F    Human Annotation Details

All human evaluations are performed using the Amazon Mechanical Turk (MTurk) platform. 218 workers from English native speaking countries, at least 5,000 HITs, and acceptance rate $\geqslant 50$, are selected based on their passing performance on a paid qualification HIT. The workers are paid with an average rate of \$15/hour. An IRB exemption was obtained for the institution's internal institutional review and ethics board, and we did not collect any denanonymizing information nor do we publish with our dataset sensitive information such as MTurk IDs.

We collect acceptability labels for critic training using the template in Figure 9. For each image and its set of annotated question, answer, rationales (QARs), we run deduplication by clustering the QAR's using hierarchical clustering[1] with their semantic similarity measured by the SentBert `paraphrase-MiniLM-L6-v2` model [13]. We select five question, answer, and rationale triples by getting the roots of the fiver clusters and considering them as the annotation candidates for each image. Using 4,800 images and 24K QAR's, we run the annotation pipeline following Section 2.3 and acquire the acceptability labels for the critic.

Figure 10 shows the template to conduct the pairwise human evaluation comparing ours to chatgpt responses with VCR questions and images [18]. To reduce the label selection bias, we randomize the order of two responses. 300 (image, QAR) pairs are selected for evaluation where there is no overlap among the images. Three annotators are asked to make a selection, and instances that did not receive at least two votes are not considered in each evaluation criteria, which we found to be 6% on average.

## G    Additional Qualitative Examples

In Figure 11, we present qualitative results of BLIP-2 FlanT5$_{\text{XXL}}$ and Mini-GP4 models trained with LSKD, for answering VCR questions [18]. The results demonstrate that both models are capable of accurately identifying the relevant person performing the action. For instance, in the first example, the models correctly identify [1] as a dentist due to the presence of a lab coat. Similarly, in the second example, they recognize that [0] is the individual talking on the phone. Notably, the Mini-GPT4 model, which employs the more powerful language model Vicuna [1], produces more precise answers. For instance, it mentions specific actions like tooth cleaning and identifies [0] as seated in the dentist's chair. Additionally, it hypothesizes that [0] might be engaged in conversation with other workers or superiors based on the observation of holding a phone. This observation suggests that LSKD benefits from employing a language model with enhanced capabilities as indicated by the human evaluation results in the main paper.

We also show failure cases in Figure 12. We observe that the models are capable of correctly identifying the individuals, such as recognizing [1] as the person wearing a black hoodie and [0] as the individual with closed eyes standing in the doorway. However, they 1) demonstrate a lack of spatial reasoning. For instance, the T5 model hallucinates that the boy is "standing on a shelf of canned food," while Mini-GPT4 hypothesizes that he would "not damage the objects" if he were to fall over, despite the close proximity of the objects in the first example. Additionally, in the second example, the models exhibit a 2) deficiency in fine-grained understanding of people's expressions. Although [0] displays a disgusted facial expression, the T5 model incorrectly interprets it as curiosity and interest, while Mini-GPT4 predicts that she is feeling nervous. These observations indicate that while the models are able to correctly identify the relevant regions, they still lack the capability for nuanced and intricate understanding that necessitates more sophisticated reasoning of visual content.

---

[1]We use the scipy library `https://docs.scipy.org/doc/scipy/reference/cluster.hierarchy`.

## H  Error Bars

We report error bars for the BLIP-2 [8] trained with LSKD in Table 2 of the main paper. We run three experiments with different random seeds and show the results in Table 3. Note all other methods are evaluated with off-the-shelf zero-shot models, hence we only report error bars just for our method.

| | VCR | | | Sherlock | VisualCOMET |
| --- | --- | --- | --- | --- | --- |
| | Q → A | QA → R | Q → AR | Comparison | Acc@50 |
| BLIP-2 ViT-G + LSKD | 58.8 ± 0.12 | 56.3 ± 0.07 | 33.2 ± 0.09 | 30.1 ± 0.09 | 40.0 ± 0.11 |

Table 3: Error bars of LSKD on zero-shot localized visual reasoning tasks (last row of Table 2).

## I  Limitations

One limitation is the recognition bottleneck of the verbalizers, in which off-the -shelf vision language models may encounter errors in object detection or action recognition. With a combination of verbalization output, the LLM largely ignores irrelevant and incoherent content in the image, but is still prone to generating erroneous data. We made efforts to mitigate the issue by training a supervised critic model on a subset of data to filter out erroneous cases. However, it should be noted that the critic model cannot guarantee the exclusion of all irrelevant instances. Despite these limitations, the trained LSKD models exhibit notable improvements and demonstrate impressive generation capabilities when applied to localized visual reasoning tasks.

Another limitation is the coverage of questions in the data. As shown in Table 2, the dataset encompasses various question types; however, there may still exist certain question categories that are underrepresented or not adequately captured (*e.g.* object counts, potential challenges, other inferences). This limitation could potentially affect the generalizability of the models trained on the dataset to specific question types that are underrepresented or absent.

**Global Descriptors**

I think this image takes place at raft, or beach.
Some objects I see are surfboard, paddle, paddle, board, and board.
There also might be a mini marcos, board short, surfboard, stand up paddle surfing, and surfboard shaper in this image.

[OFA-H Localized Narratives]
Description 1: man and woman standing. The man is holding a surfing board. In front of him a girl is there. At the background there is water and a sky. To the left side there is a surf board.
Description 2: on the right we can see a man and a woman holding a surfing board and smiling. We can see a kid walking and a surfboard at the bottom. In the background we can see sky, clouds and water.
Description 3: a woman and a man wearing shorts standing on the sand and holding a surfboard in their hands. There is a kid walking on the sand. In the background there is a sea. In the sky there are some clouds.
Description 4: a girl is running in the sand towards the man, who is holding a surfboard. On the left side, there is a surfboard. Behind the man, there is a lady, who is holding a skateboard and is smiling. In the background, there is water and there are clouds in the sky.
Description 5: a kid is standing on the sand. In front of her a person is standing and holding a skateboard. Behind them there is an ocean.

**Local Descriptors**

[BLIP-2 Region Captions]
[0] man in black shorts [(0.56, 0.3), (0.8, 0.84)]
[1] little girl in striped bathing suit [(0.22, 0.61), (0.49, 1.0)]
[2] a woman wearing a pink shirt [(0.3, 0.3), (0.44, 0.62)]
[3] sunglasses on a woman's face [(0.32, 0.33), (0.4, 0.35)]
[4] a white surfboard on the sand [(0.0, 0.73), (0.1, 0.96)]
[5] a surfboard [(0.44, 0.03), (0.69, 0.75)]
[6] ropes on the surfboard [(0.47, 0.13), (0.63, 0.19)]
[7] a small wave in the water [(0.76, 0.41), (0.78, 0.42)]
[8] a sticker on a surfboard [(0.43, 0.61), (0.5, 0.65)]
[9] the man has a beard [(0.61, 0.38), (0.65, 0.42)]

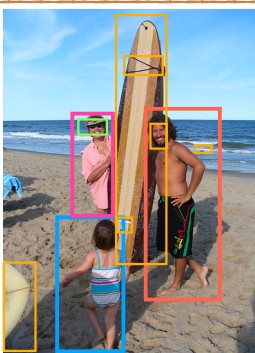

**Dynamic Descriptors**

[ChatGPT Questions + BLIP-2 Answers]
0. Question: Where does this take place at? Answer: a beach
1. Question: What is the lighting of the image? Answer: the lighting of the image is natural light
2. Question: What is the atmosphere or ambiance of the image? Answer: a family on a beach
3. Question: Does this take place inside or outside? Answer: outside
4. Question: What might be the weather like? Answer: sunny and warm
5. Question: What is the little girl wearing in the image? Answer: a bikini
6. Question: Is the sky in the background clear or cloudy? Answer: cloudy
7. Question: Can you see any ropes on the surfboard? Answer: no
8. Question: What color is the woman's shirt in the image? Answer: blue
9. Question: Is the man wearing shorts or pants? Answer: shorts
10. Question: What kind of board is on the left side of the image? Answer: a surfboard
11. Question: Is the scene taking place on a beach or at a raft? Answer: on a beach
12. Question: Are there any other ⌐ besides the man, woman, and girl in the image? Answer: no
13. Question: What is the little girl doing in the image? Answer: holding a surfboard
14. Question: Is the ocean calm or wavy in the background? Answer: wavy

Figure 5: Example of image-to-text verbalization with diverse descriptors.

Here are some specific region IDs with top-left and bottom-right bounding box coordinates normalized from 0 to 1 in [(x1, y1), (x2, y2)]:
[Local Descriptors]
======

Questions and Answers about the image:
[Dynamic Descriptors]
======

**[Instruction]**
Generate a interesting, succinct, and fun question/answer/rationale triple relating to people or objects in the scene. Select any number of person and objects referred by their ID tags (e.g. [1], [2]).
Think of what kind of interesting inference statements you can make about the people and objects.

Requirements:
- Be confident. Don't say "It's hard to tell", or "I'm just speculating"
- Do not ask about watermark or text in the bottom of the image.
- Do not ask about atmosphere, ambience, or lighting of the image.
- Do not ask about what person is wearing or the color of hair and outfit.
- Do not say the significance of what the person is wearing and their outfit.
- Do not ask question that would lead to unclear answer.
- Keep all questions/answers/rationales between 1-2 sentences.
- Only include single attribute or fact in your answer and rationale. Do not say multiple options and say "or ". For example, do not say something like "He could be a doctor or a pharmacist.", but just be confident and say "He is a doctor".
- Use the ID tags we provided in the above to refer to people or objects in your question, answer, and rationale instead of writing them out.

Try to ask something interesting or important that the viewer will be interested to know about.
These include but not limited to interesting inference, general vibe, attributes, situation, occasion, relationships of related objects.
Remember, don't mention a "description" or an "image": pretend you are actually at the scene.
Do not ask same or similar, simple question-answers already mentioned in the context.
The response should make sense when you replace IDs with their region descriptions.
Always use one or more IDs in the regions to formulate each of your response.
Now using the context, descriptions, and description about region IDs, provide **three** interesting response about people and/or objects using the region ID tags like this:

Question:
Answer:
Rationale:

Figure 6: Prompt used to generate data while referring regions by numerical IDs.

**[Global Descriptors]**
======
**Here are some specific regions with top-left and bottom-right bounding box coordinates normalized from 0 to 1 in [(x1, y1), (x2, y2)]:**
**[Local Descriptors]**
======
**Questions and Answers about the image:**
**[Dynamic Descriptors]**
======
**[Instruction]**
Generate a interesting, succinct, and fun question/answer/rationale triple about this scene. Remember to be confident, specific, interesting, and make smart connections between objects, actors, etc.

Requirements:
- Be confident. Don't say "It's hard to tell", or "I'm just speculating"
- Do not ask about watermark or text in the bottom of the image.
- Do not ask about ambiance or lighting of the image.
- Do not ask about what person is wearing or the color of hair and outfit.
- If you ask what the person is doing, be more descriptive instead of stating simple fact.
- Don't mention a "description" or an "image": pretend you are actually at the scene.
- Do not ask question that would lead to unclear answer.
- Keep all questions/answers/rationales between 1-2 sentences.
- Only include single attribute or fact in your answer and rationale. Do not say multiple options and say "or ". For example, do not say something like "He could be a doctor or a pharmacist.", but just be confident and say "He is a doctor".

Try to ask something interesting or important that the viewer will be interested to know about.
These include but not limited to interesting inference, general vibe, attributes, situation, occasion, relationships of related objects.
Remember, don't mention a "description" or an "image": pretend you are actually at the scene.
Do not ask the same or similar question-answers already mentioned in the description.
Now using the context descriptions and information about regions, provide **three** new, interesting (question, answer, rationale) response that you can infer from the context like this:

Question:
Answer:
Rationale:

Figure 7: Prompt used to generate data while referring regions by their descriptions.

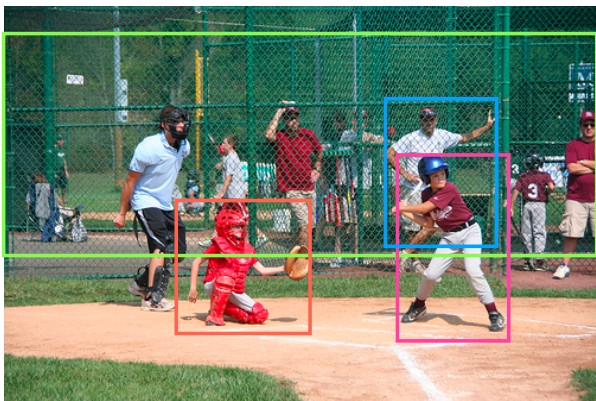

**Question**: What is the role of [3] in the game?
**Answer**: [3] is likely the catcher, responsible for catching the ball and making sure the batter doesn't steal any bases.
**Rationale**: [3] is wearing catcher's gear and is in a position on the field consistent with the role of a catcher.

**Question:** What might [0] be feeling?
**Answer:** [0] might be feeling excited or nervous.
**Rationale:** The anticipation of swinging at the ball and the pressure to perform well can cause feelings of excitement or nervousness for [0] during the game.

**Question:** Why are there people gathered behind [6]?
**Answer:** They are likely spectators watching the game.
**Rationale:** The presence of multiple people, chairs, and bags behind the chain-link fences ([6]) suggests that these individuals are gathered to watch the baseball game.

**Question**: What is the person in the white shirt doing?
**Answer**: The person in the white shirt might be a coach.
**Rationale**: The person in the white shirt seems to be standing in a position where they might be observing the game and instructing players, indicating that they could be a coach.

**Question**: What position might the player wearing the maroon shirt be playing?
**Answer**: The player wearing the maroon shirt may be a batter.
**Rationale**: The player is holding a bat and in the act of swinging, suggesting that they might be a batter.

**Question**: Is the baseball game in this image a friendly game or a competitive one?
**Answer**: The baseball game in this image is most likely a competitive one.
**Rationale**: The presence of a catcher wearing a full gear and other players wearing helmets and running around indicates that this is not just a casual game of catch.

Region IDs        Region Descriptions

Figure 8: Example of our generated corpus referring regions by IDs or descriptions.

## Overview

Thanks for participating in this HIT!

In this HIT, you will be given an **image**, **question and answer (QA)** , and **rationale** to justify the answer.
The **image** is highlighted with **tags** which is included in the text to refer to **person** or **region** in the image.
We want to determine if the **QA** is **relevant** to the image, and the **rationale** appropriately **justifies** the answer.
We also want to see if the **IDs** mentioned in text are correctly referred to the **regions** in the image (**Are they grounded correctly?**)

## Task A: QA rating

First, determine whether **just the question and answer (QA)** correctly describe the image content and the **specified regions**.
Please select from three options:
- **Accept**: QA generally delivers correct information, and it is something you can say about the image.
- **Maybe**: QA could be relevant or plausible, but we cannot confidently say that this might be true.
- **Reject**: QA doesn't make sense, or is irrelevant to any of the parts in the image. ID tags do not refer to image region correctly.

## Task B: Rationale rating

Second, determine whether the **rationale** appropriately **justifies** the answer and was **insightful** in their reasoning.
If you have selected **Reject** in **Task 1**, please select **Reject** in **Task 2** as well.
Please select from three options:
- **Accept**: Rationale justifies the answer and provides insightful reasoning.
- **Maybe**: Rationale is somewhat relevant and helpful in justifying the answer.
- **Reject**: Rationale doesn't make sense and says something irrelevant to the image. ID tags do not refer to image region correctly.

## Note:

- Please be forgiving of minor spelling and grammar errors, especially on **pronouns** (he/she vs they).
- Please **reject** if the statement includes weird, distracting artifacts such as "the description says" that are not helpful answering the question for the image.
- Regions are optionally given to help you look which part of the image is caption talking about. Feel free to ignore when making your decision.

**${qar1}**

| **Task A: QA rating** | ○ **Accept** ○ **Maybe** ○ **Reject** |
|---|---|
| **Task B: Rationale rating** | ○ **Accept** ○ **Maybe** ○ **Reject (select Reject if Reject is selected in Task A)** |

**${qar2}**

| **Task A: QA rating** | ○ **Accept** ○ **Maybe** ○ **Reject** |
|---|---|
| **Task B: Rationale rating** | ○ **Accept** ○ **Maybe** ○ **Reject (select Reject if Reject is selected in Task A)** |

**${qar3}**

| **Task A: QA rating** | ○ **Accept** ○ **Maybe** ○ **Reject** |
|---|---|
| **Task B: Rationale rating** | ○ **Accept** ○ **Maybe** ○ **Reject (select Reject if Reject is selected in Task A)** |

**${qar4}**

| **Task A: QA rating** | ○ **Accept** ○ **Maybe** ○ **Reject** |
|---|---|
| **Task B: Rationale rating** | ○ **Accept** ○ **Maybe** ○ **Reject (select Reject if Reject is selected in Task A)** |

**${qar5}**

| **Task A: QA rating** | ○ **Accept** ○ **Maybe** ○ **Reject** |
|---|---|
| **Task B: Rationale rating** | ○ **Accept** ○ **Maybe** ○ **Reject (select Reject if Reject is selected in Task A)** |

Figure 9: Template for acceptability annotation to train the critic model.

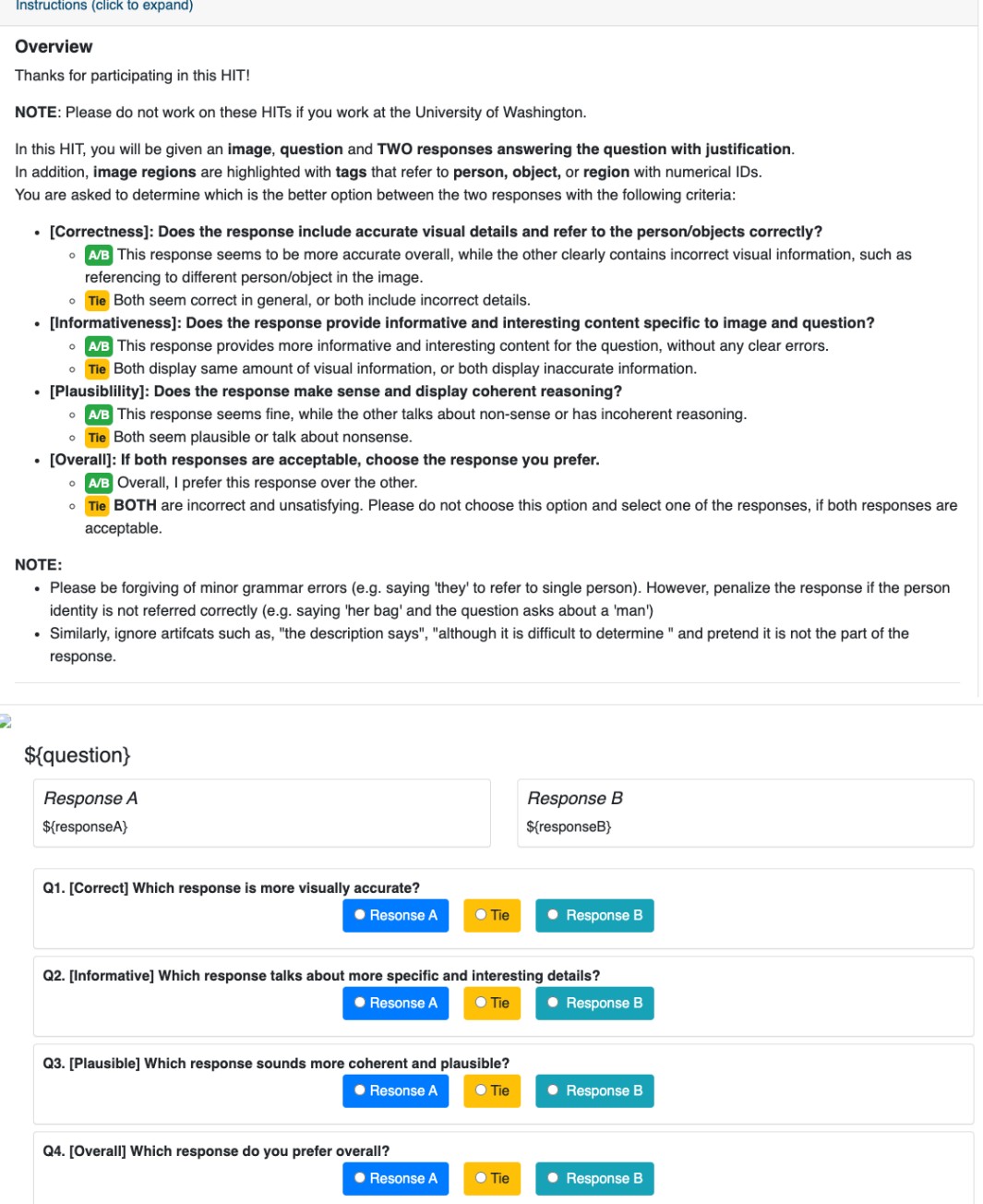

Figure 10: Template for pairwise human evaluation.

**Question**: What is [1] doing?

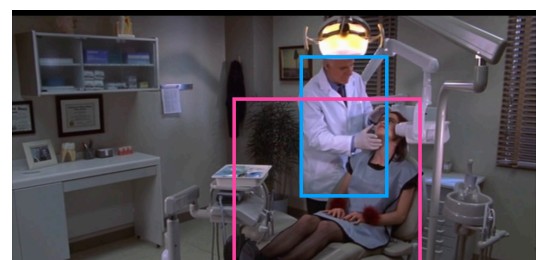

| GT | **Answer**: [1] is operating on a patient.
**Rationale**: [0] is in his clinic, with a patient on the seat as he leans over her to examine her teeth. |
|---|---|
| **LSKD**
[BLIP-2
FlanT5 XXL] | **Answer**: [1] is looking at a dental x-ray.
**Rationale:** The person is wearing a white lab coat and is standing next to the dental chair, indicating that they are a dentist or dental assistant. |
| **LSKD**
[Mini-GPT4] | **Answer**: [1] is doing a tooth cleaning.
**Rationale:** [1] is wearing a white lab coat, which suggests that he is a dentist, and [0] is sitting in a dentist 's chair, which further suggests that she is receiving a dental treatment. |

**Question:** Who is [0] talking to?

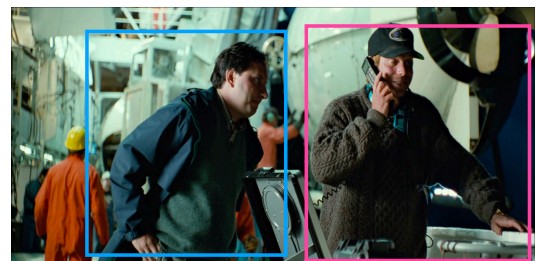

| GT | **Answer:** He's talking to someone on the phone.
**Rationale:** He has a phone up to his head and his mouth is open as if he's talking. |
|---|---|
| **LSKD**
[BLIP-2
FlanT5 XXL] | **Answer:** [0] is speaking to a person on the phone.
**Rationale:** [0] is standing on the ground and appears to be speaking to someone, indicating that they are engaged in some sort of conversation. |
| **LSKD**
[Mini-GPT4] | **Answer**: [0] is talking to a group of people in the distance, possibly fellow workers or superiors.
**Rationale**: [0] is holding a phone and standing in a group of people, suggesting they may be doing some sort of work-related communication or reporting. |

Figure 11: Qualitative Examples generated with different models trained with LSKD.

**Question**: What would happen if [1] fell backwards ?

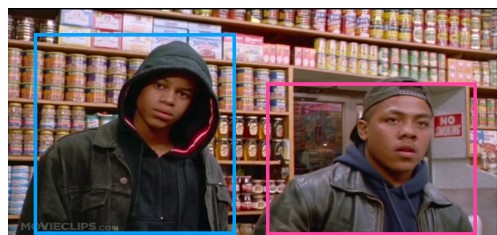

| GT | **Answer:** The jars would fall off the shelves . 
 **Rationale:** There is nothing keeping the jars on the shelves and if they fall off, they will break. |
|---|---|
| **LSKD** 
 [BLIP-2 
 FlanT5 XXL] | **Answer:** [1] could potentially hurt themselves by falling backwards as they are standing on a shelf of canned food. 
 **Rationale**: The region [1] shows that person in a black hoodie standing on a shelf of canned food with other people in the background, suggesting that they could potentially hurt themselves by falling backwards. |
| **LSKD** 
 [Mini-GPT4] | **Answer**: [1] might hit their head, but would likely not damage the objects on the shelves. 
 **Rationale**: [1] is wearing a black hoodie and standing in front of a shelf with objects, so if they were to fall backwards, they would be able to catch themselves before hitting the objects on the shelf. |

**Question**: How is [0] doing?

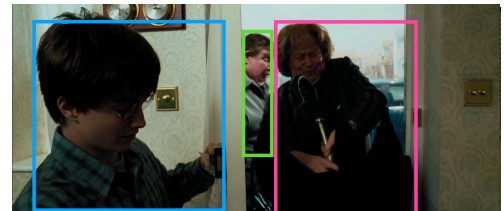

| GT | **Answer:** [0] is upset and disgusted. 
 **Rationale:** [0]'s expression is twisted in disgust. |
|---|---|
| **LSKD** 
 [BLIP-2 
 FlanT5 XXL] | **Answer:** [0] is likely feeling curious and interested, as they are standing in the doorway with their *hands raised*. 
 **Rationale**: The posture and expression of [0] suggests a high level of *curiosity* or *intrigue*, which could be related to the fact that the doorway is being held open for them. |
| **LSKD** 
 [Mini-GPT4] | **Answer:** [0] appears to be *feeling nervous*, given their closed eyes and tense body posture. 
 **Rationale:** The fact that [0] is closing their eyes and seems to be trying to *avoid eye contact* suggests that they are feeling anxious or nervous. |

Figure 12: Examples with minor errors in spatial reasoning and expression comprehension.