# OpenReview forum: "Localized Symbolic Knowledge Distillation for Visual Commonsense Models"
_NeurIPS.cc/2023/Conference — NeurIPS 2023 poster_

### Official Review · Reviewer_uG1j · 2023-06-27

**Soundness:** 2 fair
**Presentation:** 3 good
**Contribution:** 3 good
**Rating:** 5
**Confidence:** 4

**Summary:**

This paper proposes a framework that can generate global, local, and dynamic descriptions. ChatGPT can generate question-and-answer pairs containing specific regions or descriptive phrases with the proposed tricks. A critic model is trained to filter the generated data. Finally, the localized corpus is used to fine-tune the vision language model.

**Strengths:**

1. This paper proposes a way to acquire localized commonsense knowledge data and enable the vision and language model to answer localized visual commonsense questions. The workload of this work is impressive.

2. The fine-tuned vision language model achieves promising zero-shot performance for three localized visual reasoning tasks.


**Weaknesses:**

1. The proposed framework is more like engineering work. The proposed method to generate localized data pairs is straightforward, and the way to enable the local-level question-answering ability is not novel, limiting its scientific contribution.

2. There are too many tricks, including filtering by critic model and prompts for acquiring question-answering pairs, which decreases the reproducibility and robustness of the whole pipeline. Furthermore, the selection of the vision-language model and large language model, and hyperparameters for each model in each step increase the complexity of the proposed pipeline, and the final vision-language model may be sensitive to the generated data, which makes it hard to iterative improve the final model's performance by improving any step in this pipeline.

**Questions:**

This paper proposes an impressive engineering framework. It would be better to analyze the impact of the quality and amount of the generated data on the final performance of the visual language model.

**Limitations:**

I did not find a potential negative societal impact of this work.

---

> ### Author Rebuttal · Authors · 2023-08-10
>
> We thank the reviewer for acknowledging that the **workload is impressive** and **achieves promising zero-shot performance for three localized visual reasoning tasks.** We now address the reviewer's comments in the following section.
>
> **1. The proposed framework is more like engineering work. The proposed method to generate localized data pairs is straightforward, and the way to enable the local-level question-answering ability is not novel, limiting its scientific contribution.**
>
> Previous models enabled local-level commonsense question-answering ability via training on human annotation dataset such as VCR, which cannot be scaled automatically. On the other hand, we explore novel ways of automatically extracting localized visual commonsense knowledge at scale, which requires a heavy engineering workload due to their nature of automated pipeline. In fact, reviewer UzVU states that we investigate a “novel research question to distill visual commonsense knowledge”, which is “an interesting question that previous works under-explore”. We are also “the first work about automatically acquiring visual commonsense”.
>
> Reviewer Cm2R has pointed out that our framework has limited innovation compared to the original symbolic knowledge distillation. Following our response, we argue that we investigate a different problem when the teacher and student are given different input modalities, where the teacher has no access to the raw image content. This leads to a new framework of utilizing automated image descriptors to capture the raw image content interpreted by the teacher model. Different from prior multimodal generation work such as LLaVA, we also generate localized data pairs to help grounding of vision language models. Given these differences, we believe our framework presents a novel approach to localized, visual commonsense understanding.
>
> **2. There are too many tricks, including filtering by critic model … which decreases the reproducibility and robustness of the whole pipeline. …It would be better to analyze the impact of the quality and amount of the generated data on the final performance of the visual language model**
>
> Based on our empirical studies, we find that the components of verbalizers and filtering by critic model are crucial to enable the knowledge distillation.
>
> We first argue that **training the critic model is necessary to eliminate hallucinations and irrelevant generations to guarantee the dataset quality**. This is illustrated qualitatively in Figure 1 of the rebuttal where instances with incorrect visual details are given lower scores. Next, the human evaluation in Section 3 of the rebuttal clearly shows the benefits of filtered data based on their visual correctness (2.41 vs 1.97) and justification (2.34 vs 1.87).
>
> Based on these findings, we further analyze the impact of the quality of the generated data using the score from critic model.  Keeping the number of training data at 300K, we see if letting the critic model to be more critical would lead to increase in dataset quality. This can be controlled by the threshold value of critic model where we only keep the instances that score higher than the threshold.  Figure 3 shows the performance trend based on the threshold value, where we see a near-linear increase on the final performance with more aggressive filtering. This **implies that the *quality* of generated data could be controlled by the critic model**.
>
> In Table 2 of the rebuttal, we evaluate the performance of the pipeline using varying amounts of generated data, specifically comparing the results from datasets of 150K and 1M in size. We see that **increasing the scale leads to better performance in all of localized visual reasoning tasks**.
>
> We are also open to running any other analysis that reviewers would like us to investigate.
>
> **3. Are prompts for acquiring question-answering pairs necessary?**
>
> In response to Reviewer MRhi, we found that adding all three verbalizer descriptors yield the highest scores from the critic. For instance, the **adding the QA descriptor provides the biggest gain from 49.0 to 58.4**. This means that calling all the descriptors, including QA, is necessary to get more high-quality data as well.

---

> > ### Comment · Reviewer_uG1j · 2023-08-15
> >
> > I appreciate the clarification provided regarding the engineering workload and the quality of the data produced. Hence, I decide to raise my score to 5. Moreover, I recommend that the authors consider making the generated data publicly accessible to facilitate future research endeavors.

---

> > > ### Author Response · Authors · 2023-08-21
> > >
> > > We thank the reviewer for reading our rebuttal and considering re-evaluation of the score. We will provide a public link to access the generated data and reproducible code in the final version.

---

### Official Review · Reviewer_MRhi · 2023-07-01

**Soundness:** 3 good
**Presentation:** 3 good
**Contribution:** 3 good
**Rating:** 5
**Confidence:** 4

**Summary:**

This paper aims to build a instruction following model for **localized** visual commonsense reasoning. The work differs from existing works in that it is able to reason about localized image regions with boxes, without using complicated referring expressions. The paper first collects data by distilling LLMs, i.e., prompting GPT using verbalized image descriptions, which is a concatenation of 3 descriptors: global, local, and QA pairs. Then, the data collected in the first stage is filtered by a critic filter trained using human annotations. Finally, the filtered data is used to train a BLIP-2 model. Experiments are done on 3 datasets for the localized visual reasoning task, showing the performance of the proposed method.

**Strengths:**

- The paper is clearly motivated and the problem studied is well described.
- The dataset collection method is well designed. Using (global, local, QA) descriptors to prompt LLM in order to generate question, answer and rationales.
- The effectiveness of the critic filtering is well-analyzed and clearly shown in Fig 3.
- The paper is well written and easy to follow.


**Weaknesses:**

My major concern is about experiments. The current results may not be very extensive to show the effectiveness of the proposed method.
- Box size is an important factor, since the task is localized reasoning. How does the model perform on large and small objects? How does this compare to other methods? First, it would be helpful to describe the distribution of the bbox sizes in the generated data. Second, in the final evaluation, it is also helpful to study the model performance change wrt different box sizes.
- three descriptors (global, local, QA) are used to prompt LLMs. Is there a study to show the contribution of each of the three? Since QA is not as widely used as the other two, additional discussions/results showing its effectiveness would be good.
- Very limited baselines are compared with. In table 2, only CLIP is compared. In table 3, only ChatGPT. More baselines should be considered. For example, for tab 2, while zero-shot baselines are hard to find, it is still beneficial to show some non zero-shot methods (methods on the leaderboard of these datasets). Although the model performance is not directly comparable, it can give readers some sense of how well the model performs.
- (potential extension) While the contribution on the data collection/distillation part is strong, the contribution on the model part is weak. Basically, BLIP-2 is directly used and trained on the collected data. Could there be some model design specific for handling “localized” information? Moreover, the “localized” reasoning in the current paper is constrained to color-coded bbox, while more general forms like free-form masks can be a further extension.
- The work is in principle similar to LLaVA, extending it to be “localized”. Could the authors discuss the difference with LLaVA in more detail?
- The title suggests “symbolic”, how is this work related to “symbolic”? I feel this is mis-leading.
- some typos: L168 QA-MSE, L146 QAR is not defined.


**Questions:**

See weakness.

**Limitations:**

N/A.

---

> ### Author Rebuttal · Authors · 2023-08-09
>
> We thank the reviewers for acknowledging that “the dataset method collection is well designed” and “effectiveness of critic filtering is well-analyzed”. We address the concerns raised by the reviewers.
>
> **Bounding Box Size Distributions and Performance Analysis**
>
> Figure 2 in the rebuttal shows the distribution of normalized bounding box sizes in the filtered corpus, highlighting the width, height, and the area. We notice that almost 50% of the bounding boxes have the normalized area $\lt$ 0.05, suggesting that small objects are well-covered in our corpus. The height shows more uniform distribution than the width, indicating that there are more bounding boxes with smaller widths and the width mainly clusters in the range of 0.1-0.2. This reveals that the corpus contains not just large and prominent objects, but also small or narrow objects that often require attentive vision models to recognize.
>
> We use the Sherlock comparison task to study the model performance change w.r.t different bounding boxes as their dataset consists of single bounding boxes with diverse sizes. We measure the **Pearson’s correlation between the input bounding box size and the comparison accuracy**: $\mathbf{\rho}$ = -0.12 with p-value $\lt$ 0.05.
>
> Based on the correlation, we see that the performance is actually higher for smaller objects. One might indeed initially think that larger bounding boxes would result in better performance, as they could potentially encompass more features or more of the object of interest. We hypothesize that the negative correlation is due to that:
>
> - **Specificity**: Smaller bounding boxes quite often are more specific in identifying the target objects, thereby reducing the complexity of the region and making it easier for the model to focus and reason.
> - **Clutter**: Larger bounding boxes might include more "noise" or irrelevant objects/background - which could mislead the model during the reasoning process as it gets distracted by extraneous details.
>
> This is reflected in the dataset distribution as smaller objects are more covered in the dataset generation. The analysis on the bounding boxes sizes will be included in the final paper.
>
> **Contributions of Three Descriptors to prompt LLMs**
>
> We run ablation studies of the descriptor components in the ChatGPT prompt by using the critic model to score the ChatGPT generations. We collect QAR instances for 500 images and calculate the average score with higher score aligned with human preference. **We see that using all descriptors provides the best results**, and in fact the QA descriptor provides the biggest jump (from 49.0 to 58.4). We will add these results in the final version.
> - Full verbalizations: 58.4
> - No Localized Narratives: 56.1
> - No CLIP: 52.1
> - No Global Descriptions: 54.3
> - No Region Descriptions: 49.8
> - No QA: 49.0
>
> **Limited Baselines**
>
> We agree that limited baselines are compared due to difficulty of finding zero-shot baselines in the localized reasoning tasks. In Table 1 of the rebuttal, we additionally include the BLIP baseline results and the finetuned results that achieve top of the leaderboard on these dataests. The gap between finetuned and zero shot models suggests that localized visual commonsense is not present in pre-trained vision language models. This underscores the importance of our work in filling these gaps by adding localized visual commonsense to existing models.
>
> **Extension of Localization Approach**
>
> We constrain our study to color-code bounding boxes following prior works [Zellers et. al., Hessel et. al.] known to effectively encode localized information. One way to improve the handling of “localized” information is to directly encode the information as floating points in the text input and output. Another extension is using the segmentation masks over the bounding boxes for more refined localization of the desired objects and regions. We leave it as future work to improve the localization aspect.
>
> **Difference with LLAVA**
>
> - The main difference between LLAVA and our pipeline:
>   - LLAVA uses GPT-4 as the teacher model, while we use ChatGPT.
>   - LLAVA relies on human annotations of captions and object bounding boxes in COCO images, while we utilize diverse, learned descriptors to extend to images without human annotations.
>   - We train additional supervised critic to filter irrelevant generations made by the teacher model to ensure higher dataset quality, while LLAVA does not have filtering mechanism.
> - As discussed in the main rebuttal and response to Reviewer Cm2R, we compare ours with LLaVA in the localized reasoning task on the BLIP-2 ViT-G model. We see that our corpus with critic filtering gives clear improvement in various localized reasoning tasks in Figure 3 of the main paper. Table 2 in the rebuttal shows more comparisons while fixing the dataset size to be the same. We see a clear improvement of using our corpus over LLAVA in localized visual commonsense reasoning tasks with the same dataset size (row 2 and 4 in Table 2), and the gap is increased with more training data (row 3 and 4 in Table 2).
>
> **How is this work related to symbolic?**
>
> The “symbolic” terminology is introduced in [West et. al.] to denote that knowledge is distilled from the teacher to student model “symbolically as text”. Our framework is different in that we include verbalizers to prompt the teacher model to account for the modality gap as per response to Reviewer Cm2R. We will clarify this terminology in the introduction.
>
> **Typos**
>
> Thank you for pointing out the typos. We will define QAR as (Question, Answer, Rationale) and fix any remaining typos accordingly in the final version.
>
> [1]: MERLOT: Multimodal Neural Script Knowledge Models [Zellers et. al.]
>
> [2]: The Abduction of Sherlock Holmes: A Dataset for Visual Abductive Reasoning [Hessel et. al.]
>
> [3]: Symbolic Knowledge Distillation: from General Language Models to Commonsense Models [West et. al.]

---

### Official Review · Reviewer_UzVU · 2023-07-06

**Soundness:** 3 good
**Presentation:** 3 good
**Contribution:** 3 good
**Rating:** 8
**Confidence:** 4

**Summary:**

In this paper, authors argue that for multimodal LLMs, the previous input formulation is too rigid: either needs to specify the region models should focus on, or brings a verbose object description to refer to the region. A more natural referring expression strategy can help models better understand where the model should pay more attention to and the intent of the input questions. The proposed approach, called Localized Symbolic Knowledge Distillation (LSKD), aims to provide more natural referring expressions. It involves providing literal descriptions of images to a large language model and allowing the model to connect these descriptions with a holistic understanding of the scene. In addition, localized references to specific regions within the image are provided to encourage the model to generate commonsense inferences about those regions. This method effectively distills the large language model's capacity for global and local scene understanding. They demonstrate the effectiveness of this approach by achieving state-of-the-art zero-shot performance on localized visual reasoning benchmarks and conducting human evaluations.

**Strengths:**

Novel research question to distill visual commonsense: Distilling visual commonsense is an interesting question that previous works under-explore. The distilled visual commonsense could provide very useful information for understanding complex situations covered in datasets like VCR, which will make the reasoning more grounded and reliable. Also, this is the first work about automatically acquiring visual commonsense to my best knowledge.

Comprehensive experimental results on downstream tasks: I'm very glad to see the distilled knowledge could further help base models on downstream tasks like VCR, Sherlock and VisualCOMET. Also, the small models can even outperform ChatGPT with the help of LSKD when describing the useful information in localized reasoning process. Besides, authors design experiments to examine the effect of steps in LSKD generation methods, which further make the argument stronger.

**Weaknesses:**

Lack of comparison with Sherlock and VisualCOMET: Although the current experiments support that LSKD is helpful for downstream tasks, I'm still a bit confused if LSKD is better than Sherlock and VisualCOMET knowledge bases. These two datasets are both large-scale and of great quality. It's better to compare with competitive resources, instead of just showing that considering external knowledge will help.

Comparing VL models and ChatGPT is unfair: Although ChatGPT is prompted with verbalizers, it still can only accept text information, which may ignore many visual information in the images. Especially for datasets like VCR, answering questions may need a very careful observation to some detailed places, which is what VL models like BLIP-2 is better at.

Presentation in introduction section is confusing: I don't quite understand the relation between the first and second half of the introduction section. For the first half, I think the authors are trying to say a better referring expression is important. But in the second half (which is the core part of the paper), they just focus on how to distill visual commonsense knowledge. They are a bit connected because both parts mention localized reference, but it has very loose connection with discussion about the quality of referring expression.

**Questions:**

1. Is there any way to apply this method to the datasets with no bounding box annotations? That will further improve the scalability of this method.
2. Is there any study to explore the correlation between the ration of error training instances and the downstream task performance?
3. Could you also apply LSKD to models like CLIP? It seems that in Table 2, they just use BLIP-2 ViT-G to show the effectiveness of LSKD.
4. More qualitative error analysis is welcome in the modified version.

**Limitations:**

The limitations about introducing errors in the intermediate steps are reasonable.

---

> ### Author Rebuttal · Authors · 2023-08-10
>
> We thank the reviewer for positive feedback that we explore an interesting problem of localized commonsense reasoning that **previous works under-explore**, and acknowledge our novelty (**the first work about automatically acquiring visual commonsense**). The reviewer is **glad to see the distilled visual commonsense knowledge could further help base models** to make them **more grounded and reliable.**  We address the concerns raised in the weaknesses and questions.
>
> **Comparison with Sherlock and VisualCOMET**
>
> For Sherlock and VisualCOMET, we fully acknowledge the large-scale and high-quality nature of these knowledge bases, and we agree with the reviewer that comparison with them is an imperative step. As discussed in the main rebuttal, we include results based on models trained on our corpus (Row 3) with these human-annotated corpora in Table 2 (Row 5 and 6). make the following observations:
> - Ours vs Sherlock (Row 3 and 5):  Sherlock yields better results in Sherlock and VCR, and falls short in others.
> - Ours vs VisualCOMET (Row 3 and 6): VisualCOMET yields better results in VisualCOMET, and falls short in others.
>
> Not surprisingly, evaluating the same task as the existing training corpus leads to higher performance than LSKD (e.g. training on VisualCOMET corpus yields better results in VisualCOMET data than LSKD). We observe that LSKD provides benefits in terms of better generalization across diverse visual reasoning tasks. In fact, training with human authored corpus leads to considerable drop than the zero-shot models in Visual7W tasks. Such drop from zero-shot model, however, is not observed when LSKD is applied due to their diverse nature of knowledge corpus.
>
> We also like to point out that another benefit of LSKD pipeline over the human authored corpus is their annotation cost and scalability. The cost of annotating human-written knowledge corpus is higher than that of making API calls and takes significantly more time and effort if one were to scale the dataset collection.
>
>
> **Comparing VL models and ChatGPT is Unfair**
>
> We compare VL models and ChatGPT as a way to make a comparison between student and teacher models in the knowledge distillation pipeline. Despite the modality gap, we still consider ChatGPT with verbalizers a strong baseline, since the descriptors inform place information, specific descriptions of regions, and dense QAs to inform the visual content to help to answer the question. Prior works show that language models can display strong zero-shot visual reasoning [Zeng et. al., Zhu et. al., Wu et. al., Lu et. al.]  by utilizing the text descriptions from vision tools.  In response to Reviewer MrHi, we show that small and narrow objects are covered by the region proposal pipeline, suggesting that descriptors can cover specific details of the image. However, we do agree that there remains an information bottleneck in the ChatGPT evaluation when understanding the visual content and will address these concerns in the limitation section.
>
> **Introduction doesn't flow as well**: Thank you for the comment! We will fix the introduction to fit the flow between the need for localized visual commonsense models and distilling visual commonsense knowledge.
>
> **Apply to datasets with no bounding box annotations**: Yes! Please refer to the main rebuttal and updated results in Table 1. We see consistent improvements on datasets with no bounding box annotations (AOKVQA, SNLI-VE, and Visual7W).
>
> ** Any correlation between training instances and downstream performance?**: In Figure 3 of the main paper, we find that there is a positive correlation between the filtering threshold value as data quality control and the downstream task performance. In other words, making this critic model more critical helps filter quality knowledge statements from large quantities (L233-235). Thus, even from extracting the generations with the same teacher model, we observe that decreasing the error of training instances with the critic filtering is crucial to see gains in downstream task performance.
>
> **LSKD to other models**: We thank the reviewers for the suggestion, and agree that LSKD can be applied to models like CLIP. We run experiments mainly on BLIP-2 ViT-G due to their superior zero-shot performance than CLIP. Due to limited time and computation resources in the rebuttal period, we have not included their results but will do so in the later discussions.
>
> **More qualitative error analysis**: In addition to Figure 4 of the main paper, we will add more qualitative analysis in the main paper.
>
> - [1]: Socratic Models: Composing Zero-Shot Multimodal Reasoning with Language [Zeng et. al.]
> - [2]: ChatGPT Asks, BLIP-2 Answers: Automatic Questioning Towards Enriched Visual Descriptions [Zhu et. al.]
> - [3]: Visual ChatGPT: Talking, Drawing and Editing with Visual Foundation Models [Wu et. al.]
> - [4]: Chameleon: Plug-and-Play Compositional Reasoning with Large Language Models [Lu et. al.]

---

> > ### Comment · Reviewer_UzVU · 2023-08-19
> >
> > Thanks for answering the questions! I'm glad to see most of them are addressed! And I hope that they can be discussed in the following version. I will raise my score a bit.

---

### Official Review · Reviewer_Ta54 · 2023-07-07

**Soundness:** 3 good
**Presentation:** 3 good
**Contribution:** 3 good
**Rating:** 5
**Confidence:** 4

**Summary:**

The paper introduces Localized Visual Commonsense models that enhance vision-language (VL) models by allowing users to specify specific regions within images. The authors train their model by collecting commonsense knowledge from a large language model (LLM) using global literal image descriptions and automatically generating local literal region descriptions from VL models. They also use a critic model to select high-quality examples. The results show that training on this localized commonsense corpus improves the performance of VL models of passing generated referring expressions to an LLM.

**Strengths:**

+ The paper is well-motivated. The introduction of Localized Visual Commonsense models represents a novel solution to build a more general multi-modal model.

+ The constructed data is of large scale, which contains 1M samples over 250K images.

+ The extensive experiments show that the proposed dataset can support model to perform better zero-shot inference.

**Weaknesses:**

- How does this Dataset reflect knowledge? Since ChatGPT's input is based on factual descriptions of images, how can we ensure the questions generated by ChatGPT are related to external knowledge? On the other hand, how do the authors define what are knowledge-related questions?

- For some knowledge-related questions, it seems that ChatGPT needs to guess the state of the subject in the image. Could this process result in hallucination issues? I notice that there's a critic filtering in the method, which rates the results by training on annotated data. What types of patterns is this model expected to learn, and under what circumstances does it consider the generated data to be inadequate? From Section 2.3, it seems to be judging visual correctness, but how can the correctness of the generated knowledge be determined?

For example, in Fig1, the model assumes that [0] and [2] are teaching [1]. This is a guess made by ChatGPT, as no similar descriptions are generated by the visual models, indicating that they cannot capture this information. So how would the critic model determine if this kind of guess is based on reality or is just a hallucination?

**Questions:**

My major concern with this work is how to guarantee the dataset quality since there are a lot of automatic generation methods in dataset construction. So I would like to hear more illustrations about that.

---

> ### Author Rebuttal · Authors · 2023-08-10
>
> We thank the reviewer for the feedback and valid concerns regarding how to guarantee the dataset quality, which we address below.
>
> **1. How does this Dataset reflect knowledge? How do the authors define what are knowledge-related questions?**
>
> In our work, we investigate extracting commonsense knowledge from LLMs that involves reasoning about everyday objects and concepts to make judgments and predictions about new situations. Our contribution is to extend this paradigm to handle and extract rich knowledge for the multimodal data. We do this by providing factual descriptions of objects in the image, and utilizing the LLM’s commonsense reasoning capabilities to extract complex relations and interactions among objects in the image. We hypothesize that ChatGPT can perform such tasks with localized information based on the findings from previous work:
>
>  - Wu et. al. demonstrate that ChatGPT can solve challenging zero-shot multimodal tasks via sophisticated use of prompting and diverse vision experts, including image captioning and object detection models. These tasks include spatial/coordinate understanding (can provide valid localized information) and open-world concept understanding in the multimodal domain.
>  - Bang et. al. perform evaluation of ChatGPT in commonsense reasoning and find that “ChatGPT does quite well not only in terms of answer accuracy but also in generating reasonable reasoning procedures to support its answer”.
>
> These findings suggest that **one can indeed prompt LLMs to create commonsense, inference statements with reasonable reasoning procedures in the multimodal domain, if the appropriate descriptions of the images are provided**. Similarly, our dataset collection procedure follows this intuition: 1) The global, local descriptors and the QAs by themselves give a relatively surface-level understanding of the objects and concepts. 2) The LLMs can utilize the information to probe complex relations and interactions among the objects in the form of Question, Answer, and Rationale explaining the reasoning procedures (QAR).
>
> This form of knowledge is in fact illustrated in Figure 7 of the supplemental, where we observe:
> - Localized descriptions of a man and woman standing next to each other and holding a surfboard.
> - Description of a girl is running in the sand towards the man, who is holding a surfboard.
> - Then, ChatGPT uses commonsense reasoning that the three hold close, family-like relationships, and predicts the father might be interested in teaching the girl how to surf.
>
> To empirically validate if the dataset contains relevant knowledge-related questions, we have trained the state of the art zero-shot model on the localized corpus and evaluated on downstream tasks known to require localized, visual commonsense knowledge to achieve good performance (VCR, Sherlock, VisualCOMET). The results in Table 2 in the main paper and Table 1 in the rebuttal show that the model with LSKD provides improvements on various localized and reasoning based downstream tasks, suggesting that **ChatGPT is indeed capable of generating relevant knowledge of visual commonsense understanding**
>
>
> **2. How to guarantee the dataset quality since there are a lot of automatic generation methods in dataset construction? I would like to hear more illustrations about that.**
>
> As pointed out by the reviewer, the **dataset quality is controlled by the supervised critic model trained to filter out irrelevant instances, including hallucination**. The main rebuttal reports empirical benefits of filtering and includes human evaluation of filtered data, which we summarize here:
>
> We first argue that increasing the dataset quality should be transferred to their respective downstream performance. In Figure 3, we show empirical benefits of the critic-filtering mechanism on the downstream tasks. We see that the performance strictly increases by applying the critic model than without (threshold of 0 in the left side of the graph). Applying more strict threshold on the critic filter leads to near-linear increase in performance in downstream tasks. This suggests that the dataset quality can be controlled with the filtering mechanisms using the supervised critic model.
>
> In Section 3 of the main rebuttal, we run human evaluations of instances with and without filtering based on visual correctness of QA pairs and justification of the rationale. We see that the data filtered by the critic model achieve higher ratings in both criteria by a large margin.
>
> **3. What types of patterns is the critic model expected to learn, and under what circumstances does it consider the generated data to be inadequate?**
>
> The critic follows the patterns of human annotations on evaluating the generated QARs from ChatGPT. The criteria is based on the visual correctness of QAs and appropriate justification of rationales as described in $\S$2.3. Unlike ChatGPT, the critic model is a vision language model with access to the image input to determine to eliminate cases of hallucination of objects and actions. By training on human labels, we hypothesize that the critic model is equipped with adequate visual discriminability to perform the appropriate filtering.
>
> We show an illustration of the critic model prediction in Figure 1 of the main rebuttal. The critic model uses its visual understanding skills to give high-scores instances with correct visual information (First example: gold hook in region [4] is localized correctly with appropriate description of its significance; Second example: the tag in region [5] correctly indicates the brand of the stuffed animal). We see that instances with visually incorrect and hallucinated details (bear holding another toy, scowling expression of brown teddy bears) are given low scores.
>
> [1]: A Multitask, Multilingual, Multimodal Evaluation of ChatGPT on Reasoning, Hallucination, and Interactivity [Bang et. al.]
>
> [2]: MM-REACT: Prompting ChatGPT for Multimodal Reasoning and Action [Wu et. al.]

---

### Official Review · Reviewer_Cm2R · 2023-07-07

**Soundness:** 3 good
**Presentation:** 3 good
**Contribution:** 3 good
**Rating:** 6
**Confidence:** 4

**Summary:**

This work presents a localized visual common sense model that can support region-level inputs for knowledge reasoning. Specifically, the proposed methods prompts large language models to collect commonsense knowledge from global image descriptions and local descriptions. A critic classifier trained over a small number of annotation data is used to select high-quality examples. The distilled localized common-sense corpus can be further used for fine-tuning vision-language models. The experimental results demonstrate improvement over baselines in the zero-shot setting.


**Strengths:**

a). Strong motivation; supporting region-level references and reasoning is important for many applications.

b). Technical sound;  provide detailed and clear explanation of each component; experiments and ablation studies are comprehensive.

c). Thorough evaluation; have human evaluations comparing zero-shot performance of localized reasoning with generative models.


**Weaknesses:**

a). A new way of distilling vision-language knowledge but no framework-level innovation compared with text only symbolic distillation.

b). The effectiveness of supervised critic: as the validity of instance annotation is only 45% and the trained classifier can only achieve 0.66 on F1 score. It is unclear whether the filtering process can be effective in selecting high-quality examples.


**Questions:**

a). Can the collected 1M instances of localized commonsense knowledge be used for other relevant vision-language tasks? Will the knowledge further benefit those tasks?

b). In the Section of “effect of filtering”, the comparison with LLaVa datasets seems unfair since the number of LLaVa is fixed. What is the number of training data for LLaVa?


**Limitations:**

n.a.

---

> ### Author Rebuttal · Authors · 2023-08-09
>
> We thank the reviewer for agreeing that support for **region-level references and reasoning is important for many applications**, and our experiments and ablations are comprehensive that include human evaluations of generative models. We address the comments raised by the reviewer below.
>
> **1. A new way of distilling vision-language knowledge but no framework-level innovation compared with text only symbolic distillation.**
>
> Text only symbolic knowledge distillation and its variants assume that the teacher LLM (e.g. GPT-3, ChatGPT) and student models have access to the same text input modalities to perform the knowledge distillation process. In comparison, our framework explores when the teacher and student are provided with varying levels of input modalities in that the teacher LLM does not support raw image as input, while the student multimodal model is not subjected to the same constraint. To deliver the raw image content interpreted by the teacher model, we use a range of automatic image descriptors to verbalize images as text modalities. Specifically, we introduce three types of descriptors (Global, Local, QAs) all of which have proven significant as per the response to Reviewer MRhi. We assert that this unique aspect of our framework is an innovative contribution that will enable the multimodal community to comprehensively understand effective strategies for distilling visual-language knowledge from LLMs using automated, partial image descriptions.
>
> **2. The effectiveness of supervised critic: as the validity of instance annotation is only 45% and the trained classifier can only achieve 0.66 on F1 score**.
>
> In Line 170-178, we explore tuning thresholds for the critic model and keeping only the instances whose predictions are higher than the threshold value. Figure 2 shows that the model precision increases by increasing the threshold value. Specifically, the acceptability jumps 45% to 70% by setting the threshold as 0.8. This shows that applying the critic model alone can increase the dataset quality measured by humans.
>
> In Figure 3, we show empirical benefits of the critic-filtering mechanism on the downstream tasks. We see that the performance strictly increases by applying the critic model than without (threshold of 0 in the left side of the graph).
>
> Lastly, we report human evaluation of filtered and unfiltered dataset in the main rebuttal. We show that humans prefer dataset with critic filtering based on visual correctness and justification by a large margin.
>
> **3. Can the collected 1M instances of localized commonsense knowledge be used for other relevant vision-language tasks? Will the knowledge further benefit those tasks?**
>
> Table 1 in the rebuttal shows the benefits of localized commonsense knowledge in other visual-reasoning tasks such as AOKVQA, SNLI-VE, and Visual7W. We see clear improvement from applying BLIP-2 with LSKD over BLIP-2 in their zero-shot performance. We will include these downstream task results in the final version.
>
> **4. In the Section of “effect of filtering”, the comparison with LLaVa datasets seems unfair since the number of LLaVa is fixed. What is the number of training data for LLaVa?**
>
> The LLaVA dataset consists of 150K instructions on question-answers generated by GPT-4, while ours is generated by ChatGPT. In Figure 3 of the main paper, we have sampled the training data after filtering so that the number of instances is always fixed to be 300K (L236) and make comparison with the LLaVA.
>
> To make a fair comparison with the LLaVA, we experiment with 150K training samples of our corpus with and without filtering. In Table 2 of rebuttal, we observe that **LSKD + filtering wins over training on the LLaVA corpus on localized reasoning benchmarks (VCR, Sherlock, and VisualCOMET), despite using the weaker teacher model**.  The results suggest that our creation of a new localization corpus is crucial to support the model with grounded reasoning. On the other hand, LLAVA wins on QA-based reasoning tasks as they are aligned with the nature of training corpus. We thus observe that the appropriate application of these corpuses can be task-dependent, and adopting a selective approach towards choosing the right toolset may result in significantly enhanced performance across various benchmarks.

---

> > ### Comment · Reviewer_Cm2R · 2023-08-17
> >
> > Thanks for answering the questions, which some of them are clear. But for Q2, I understand that improving the threshold will definitely increase the quality. However the acceptability is still relatively low (70%) and I am asking whether such acceptability score is enough for high-quality data filtering.

---

> > > ### Author Response · Authors · 2023-08-21
> > >
> > > We appreciate the reviewer for reading the rebuttal and providing a further clarifying question.
> > >
> > > Regarding Q2, we consider that 70% acceptability still represents a substantial enrichment of our dataset with high-quality examples. We believe the best way to verify this is **via showing a clear empirical gain** by training on the dataset with and without critic filtering, as discussed in the rebuttal to Cm2R.
> > >
> > > In **Table 2 of the main rebuttal**, we compare model performances trained on our filtered dataset versus those trained on well-established, high-quality, human-annotated visual knowledge corpora, such as Sherlock and VisualCOMET. We demonstrate our model's ability to perform at par, if not better, on localized reasoning tasks, and our dataset also provided better generalization to other visual reasoning tasks (AOKVQA, SNLI-VE, V7W). These empirical outcomes support our assertion that a 70% acceptability score can indeed produce a high-quality dataset.
> > >
> > > We acknowledge that there's room for improvement in the data filtering process to increase this acceptability score further. Future work can focus on enhancing the visual recognition capacity of the critic model to achieve this. Your noteworthy feedback will guide our future efforts in this direction.

---

### Author Rebuttal · Authors · 2023-08-09

We thank the reviewers for positive comments acknowledging that our work supporting region-level reasoning is "important and interesting tasks for many applications" (reviewer Cm2R), and "represents a novel solution to a more general multi-modal model" (reviewer Ta54). Reviewer UzVU notes that "this is the first work about automatically acquiring visual commonsense" that is "well-designed" (reviewer MRhi) and "workload is impressive" (reviewer uG1j).

We now address some common concerns shared by the reviewers.

**1. Generalization of localized reasoning corpus to other visual reasoning tasks**

In response to Reviewer Cm2R and UzVU, we measure the effectiveness of the localized knowledge corpus on other vision-language tasks not limited to datasets with no bounding box annotations. We specifically focus on ones that require high-level reasoning that would benefit from visual commonsense corpus:
  - AOKVQA [A]: that require outside world-knowledge to answer questions evaluated on multiple choice
  - SNLI-VE [B]: inference based visual entailment to test fine-grained image understanding
  - Visual7W [C]: Visual QA with focus on visual grounding and evaluations carried out on TellingQA.

Table 1 in the rebuttal displays the results of zero-shot and fine-tuned models on the considered visual reasoning tasks. We see that applying LSKD to BLIP-2 which is the best zero-shot model provides empirical improvement in every downstream tasks. This suggests that our corpus can be extended to improve on various reasoning tasks, not limited to localized reasoning, and further strengthens our contribution of corpus creation.

[A]: A-OKVQA: A Benchmark for Visual Question Answering using World Knowledge [Marino et. al.]

[B]: Visual Entailment: A Novel Task for Fine-Grained Image Understanding [Xie et. al.]

[C]: Visual7W: Grounded Question Answering in Images [Zhu et. al.]

**2. Ablation studies of localized corpus and comparison with existing, related corpus**

Several reviewers have requested to quantify the effect of filtering and dataset size, and make comparison of our localized commonsense knowledge (LCK) corpus with related resources.

***Effect of Filtering on Downstream Performance***

In Figure 3 of the main paper, we have presented the effect of data quality controlled by filtering threshold set by the critic model. We keep the training dataset as 300K and sample knowledge statements scored higher than the threshold by the critic. We found that models trained without filtering (indicated by a threshold value of 0) always performed worse than filtered datasets at any varying threshold for downstream tasks. We also see near linear trend based on the threshold value, in which **increasing the dataset quality based on the supervised critic score can lead to increase in downstream performance**.

***Comparison with existing corpus***

The LLaVA dataset consists of 150K instructions on question-answers, summary, and conversations generated by GPT-4 with no supervised critic. In **Table 2 of the rebuttal**, we report downstream results of BLIP-2 ViT-G trained with our localization corpus and LLAVA. To keep the comparison fair with LLAVA, we include results with same number of training data (150K). We notice that training on LLAVA corpus yields far less results on the localized reasoning benchmarks (VCR, Sherlock, VisualCOMET) than our subset of corpus. This is expected due to lack of support of localized reasoning in the LLAVA data. On the other hand, LLAVA provides benefits on other visual reasoning tasks, suggesting that the choice of corpus should be tailored to the specific task to optimize performance. We also notice that the critic filtering is again beneficial across all the tasks in the smaller training subset.

Next, we compare with existing human-annotated visual commonsense knowledge corpus (reviewer MRhi). In Table 2 of the rebuttal, we observe that training on human annotated corpus vastly improves the performance of their relative tasks (e.g. training on VisualCOMET boosts the VisualCOMET performance from 39.0 to 50.2). However, we notice that this could lead to worse results than other visual reasoning tasks than the zero-shot counterpart (drops from 77.1 to 70.1/70.8 in Visual7W). This suggests that human-designed datasets may lead to less task generalization due to their lack of diversity, while such trend is not observed in LSKD.

**3. Ensuring Dataset Quality**

Figure 1 in the rebuttal shows qualitative examples to understand the patterns of critic model in distinguishing good and bad examples. We see that the model mostly relies on incorrect visual details (highlighted as red) lower than the correct instances. The third instance does not have glaring visual errors but are scored lower due to statement of "largest and most prominent regions", which is ambiguous but close to false. The critic model displays good calibrations in ordering the instances, such as giving the lowest score to the instance with the most visual error.

**Human Evaluation of Filtered Dataset**

We additionally run human evaluation to measure the acceptability of data with and without filtering. We collect 500 instances the same way critic model labels are collected in Sec 2.3: 1) is the QA visually correct? and 2) does the rationale justify the answer? Likert scores from [1-3] are calculated for each criteria (higher the better).

|                | With Filtering (threshold=0.8) | Without filtering |   |   |
|----------------|----------------------|-------------------|---|---|
| QA Correctness | 2.41 $\pm$ 0.74      | 1.97 $\pm$ 0.84   |   |   |
| Rationale Justification     | 2.34 $\pm$ 0.82      | 1.87 $\pm$ 0.87   |   |   |
|                |                      |                   |   |   |

The results further suggest that applying filtering ensures higher dataset quality not only measured by downstream task performances, but also confirmed by human judgment.

---

### Comment · Area_Chair_Duph · 2023-08-20
**Thanks for your response -- AC Comment**

Dear Authors,

Thanks for providing your responses to the reviewers' comments. They are comprehensive and answer many of the concerns raised during the initial review phase. The discussion with other reviewers also provides additional context for the questions raised during the initial review. While there is still time in the discussion phase, the reviewers can engage for further clarifications.

-- Your AC

---

### Decision · Program_Chairs · 2023-09-21

**Decision:**

Accept (poster)

**Comment:**

The work received primarily positive reviews from 4 reviewers, who found the idea novel and interesting. Questions were raised during the review period regarding data quality, generalization capabilities, ablation studies, etc., which were effectively addressed in the authors' responses to some extent. The authors are highly encouraged to incorporate details from the rebuttal into the final version for completeness.